# Hyperconfined bio-inspired Polymers in Integrative Flow-Through Systems for Highly Selective Removal of Heavy Metal Ions

Masaki Nakahata [1,2] ✉, Ai Sumiya[2], Yuka Ikemoto[3], Takashi Nakamura [4], Anastasia Dudin[5], Julius Schwieger[5], Akihisa Yamamoto [6,7], Shinji Sakai[2], Stefan Kaufmann[5] & Motomu Tanaka [5,6] ✉

Access to clean water, hygiene, and sanitation is becoming an increasingly pressing global demand, particularly owing to rapid population growth and urbanization. Phytoremediation utilizes a highly conserved phytochelatin in plants, which captures hazardous heavy metal ions from aquatic environments and sequesters them in vacuoles. Herein, we report the design of phytochelatin-inspired copolymers containing carboxylate and thiolate moieties. Titration calorimetry results indicate that the coexistence of both moieties is essential for the excellent $Cd^{2+}$ ion-capturing capacity of the copolymers. The obtained dissociation constant, $K_D$ ~ 1 nM for $Cd^{2+}$ ion, is four-to-five orders of magnitude higher than that for peptides mimicking the sequence of endogenous phytochelatin. Furthermore, infrared and nuclear magnetic resonance spectroscopy results unravel the mechanism underlying complex formation at the molecular level. The grafting of 0.1 g bio-inspired copolymers onto silica microparticles and cellulose membranes helps concentrate the copolymer-coated microparticles in ≈3 mL volume to remove $Cd^{2+}$ ions from 0.3 L of water within 1 h to the drinking water level (<0.03 μM). The obtained results suggest that hyperconfinement of bio-inspired polymers in flow-through systems can be applied for the highly selective removal of harmful contaminants from the environmental water.

Access to safe water, sanitation, and hygiene is a fundamental requirement for the well-being of our society, and the sustainable development goals (SDGs) established by the United Nations involve ensuring access to these essential components of life[1]. The removal of hazardous contaminants from drinking water is critical because they adversely affect on various organisms, even at low concentrations. Therefore, the development of robust water treatment materials that selectively remove contaminants, such as heavy metal ions from aquatic environments is desirable[2,3]. Currently, heavy metal ions are removed using various materials, such as synthetic/natural zeolites, reverse osmosis membranes, and ion exchange resins[4,5]. These materials offer an efficient and cost-effective solution to treat large amounts of wastewater, but they suffer from fundamental drawbacks. The loading capacity of

[1]Department of Macromolecular Science, Graduate School of Science, Osaka University, Osaka 560-0043, Japan. [2]Department of Materials Engineering Science, Graduate School of Engineering Science, Osaka University, Osaka 560-8531, Japan. [3]Japan Synchrotron Radiation Research Institute (JASRI) SPring-8, Hyogo 679-5198, Japan. [4]Institute of Pure and Applied Sciences and Tsukuba Research Center for Energy Materials Science (TREMS), University of Tsukuba, Ibaraki 305-8571, Japan. [5]Physical Chemistry of Biosystems, Institute of Physical Chemistry, Heidelberg University, Heidelberg 69120, Germany. [6]Center for Integrative Medicine and Physics, Institute for Advanced Study, Kyoto University, Kyoto 606-8501, Japan. [7]Present address: Interdisciplinary Theoretical and Mathematical Sciences Program (iTHEMS), RIKEN, Saitama 351-0198, Japan. ✉e-mail: nakahata.masaki.sci@osaka-u.ac.jp; tanaka@uni-heidelberg.de

the currently used zeolites and ion exchange resins is typically <1 mmol per 1 g of materials. It is difficult to increase the capacity as it is determined by the area–volume ratios. Notably, the materials used for real water treatment are not highly selective, because the ions possessing similar sizes and charges are equally captured. The typical dissociation constants ($K_D$) of ion exchange resins to transition metal ions are in the order of ~$10^{-3}$ M, and the selectivity coefficient values for the metal ions are in the range of $10^{-1}$–$10^{2}$[6–8].

For the highly efficient and selective removal of heavy metal ions, learning from biological systems is a promising strategy. Plants have a highly abundant protein, phytochelatin, oligomers of glutathione that selectively capture and detoxify heavy metal ions present in soil, groundwater, and air[9,10]. The phytochelatin synthase gene (PCS) is a highly conserved gene present in all eukaryotes and several prokaryotes[11]. The process of environmental remediation using plants, known as phytoremediation, has already been put into practical use. The application prospects of this process are growing every year[12]. As shown in Fig. 1a, phytochelatin proteins form complexes with $Cd^{2+}$ and sequester them in vacuoles, while the interaction with essential ions is negligible.

We designed and synthesized polyacrylic-acid-based copolymers inspired by plant phytochelatin and examined their ability to treat water by integrating these bio-inspired polymers into a flow-through system. Similar to endogenous phytochelatins containing –COOH as well as –SH side chains, our copolymer materials consisted of poly(acrylic acid)s (PAA) that were partially functionalized with cysteine side chains. The synthesized copolymers were named pAA–Cys5 (Fig. 1b). The affinity of pAA–Cys5 for $Cd^{2+}$ was assessed by using isothermal titration calorimetry (ITC), and the structures of the polymer–ion complexes were analyzed using Fourier transform infrared (FTIR) and nuclear magnetic resonance (NMR) spectroscopy. Results from batch adsorption tests revealed that pAA–Cys5 could be used to achieve a high loading capacity that was comparable to the maximum loading capacity reported to date. By grafting of bio-inspired copolymers onto silica microparticles and cellulose membranes, 0.1 g copolymers can be confined in ≈ 3 mL volume. Moreover, the combination of the microparticles and cellulose membranes functionalized with the copolymer brushes enables the treatment of 0.3 L water to the drinking water level within 1 h.

## Results

### Material design

Figure 1 presents the strategy followed to synthesize materials inspired by biology. In plants, phytochelatins form complexes with intracellular heavy metal ions such as $Cd^{2+}$, which are isolated and stored in vacuoles (Fig. 1a). Inspired by the chemical structure of phytochelatin, a series of pAA-based copolymers (pAA–Cys5) possessing –COOH as well as –SH side chains were synthesized (Fig. 1b)[13–15]. In brief: pAA–Cys5 was synthesized by reversible addition–fragmentation chain-transfer (RAFT) radical polymerization of S-trityl-cysteine acrylamide (S-Tri-Cys-AAm) and acrylic acid (AA) using a radical initiator and a chain transfer agent, followed by deprotection of trityl group with trifluoroacetic acid (TFA). More details can be found in Supplementary Figs. 1–2.

The copolymers were synthesized following the reversible addition–fragmentation chain-transfer (RAFT) radical copolymerization of acrylic acid (AA) and AA coupled to cysteine. Supplementary Fig. 1 shows preparation of an S-protected cysteine monomer (S-Tri-Cys-AAm). A scheme for the synthesis of pAA–Cys5 is shown in Fig. S2a. Azobisisobutyronitrile (AIBN) was used as the radical initiator. Analysis of the $^1$H NMR spectra (Supplementary Fig. 2b) confirms successful synthesis. A gel permeation chromatography (GPC) chart for pAA–Cys5 (Fig. S2c) indicated a unimodal peak. The weight-average

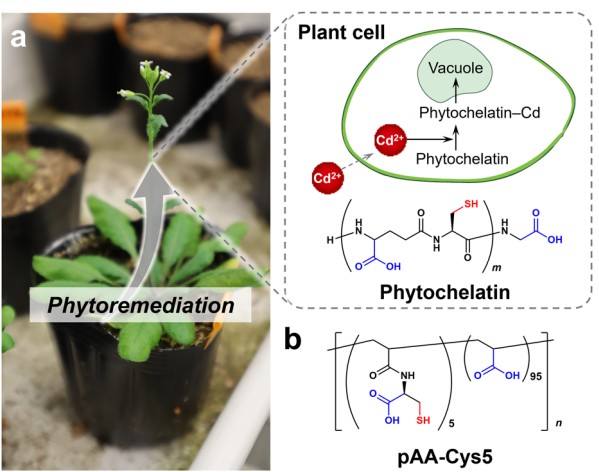

**Fig. 1 | Design of copolymer (pAA–Cys5) inspired by plant phytochelatin. a** Schematic of the process of phytoremediation occurring in plant cells. The phytochelatin forms complexes with heavy metal ions such as $Cd^{2+}$ to sequester the ions in vacuoles. **b** Chemical structure of the designed phytochelatin-inspired synthetic polymers (pAA–Cys5).

molecular weight ($M_w$) value was recorded to be $1.7 \times 10^4$, and the molecular weight distribution ($M_w/M_n$) value was 2.2.

### Validation of design of bio-inspired polymers

To verify the applicability of the design strategy in the synthesis of plant phytochelatin-inspired materials, we determined the binding affinity of pAA–Cys5 to $Ca^{2+}$ and $Cd^{2+}$ ions by measuring ITC in solution and compared the affinity with those of phytochelatin and other materials. The additional thermal power (d$Q$/d$t$) and enthalpy ($\Delta H$) plotted against the molar ratio of the cysteine side chains are presented in Fig. 2a. The global shape of the ITC data characterized by a peak ($\Delta H > 0$) indicated the coexistence of exothermic and endothermic processes. The exotherm can be interpreted as the binding of the divalent cations to the negatively-charged pAA–Cys5 brushes, whereas the endotherm represents the entropic changes caused by the release of counterions and/or the dehydration of pAA–Cys5[16–18]. The ITC data for pAA–Cys5 were fitted using a two-site model (solid lines). The $K_D$ of pAA–Cys5 to $Cd^{2+}$ ions, $K_{D(Cd)} = 2.1 \times 10^{-9}$ M per molecule, is by four orders of magnitude smaller than that to $Ca^{2+}$ ions, $K_{D(Ca)} = 2.5 \times 10^{-5}$ M per molecule, indicating that the affinity of pAA–Cys5 for $Cd^{2+}$ is significantly high. In contrast, the binding affinities of pAA (devoid of cysteine side chains) to $Ca^{2+}$ and $Cd^{2+}$ ions are too low to calculate the $K_D$ values from the ITC data (Supplementary Fig. 3). Although the direct comparison of the $K_D$ values determined by different techniques and/or under different measurement conditions is difficult, the binding affinity of pAA–Cys5 to $Cd^{2+}$ ions is significantly higher than those of previously reported materials. For example, Chekmeneva et al. determined the $K_D$ values of phytochelatin analog (($\alpha$Glu-Cys)$_4$-Gly) and glutathione by ITC measurements in 20 mM Tris-HCl buffer containing 0.1 M NaCl (pH 7.4), $3.2 \times 10^{-7}$ M per molecule and $2.0 \times 10^{-5}$ M per molecule, respectively[19]. Cheng et al. measured absorption spectra in 0.1 M Tris-HCl (pH 7.4) and obtained a comparable $K_{D(Cd)} = 3.2 \times 10^{-7}$ M for the same phytochelatin analog, ($\alpha$Glu-Cys)$_4$-Gly[20]. Visvanathan et al. synthesized random peptides mimicking phytochelatin, oligo(L-Glu-co-L-Cys), and calculated a much weaker affinity to $Cd^{2+}$ ions from the absorption spectra measured in 0.1 M Tris-HCl (pH 7.4), $K_D = 8.6 \times 10^{-4}$ M per molecule[21]. In this study, the $K_D$ of pAA-Cys5 to $Cd^{2+}$ ions ($K_{D(Cd)} = 2.1 \times 10^{-9}$ M per molecule) was determined in 10 mM Tris-HCl (pH 7.4). This value is two to five orders of magnitude smaller than the other materials including phytochelatin (Supplementary Table 1), indicating that bio-inspired synthetic pAA-Cys5 can overtake the function of the naturally occurring peptides (Glu-Cys)$_n$.

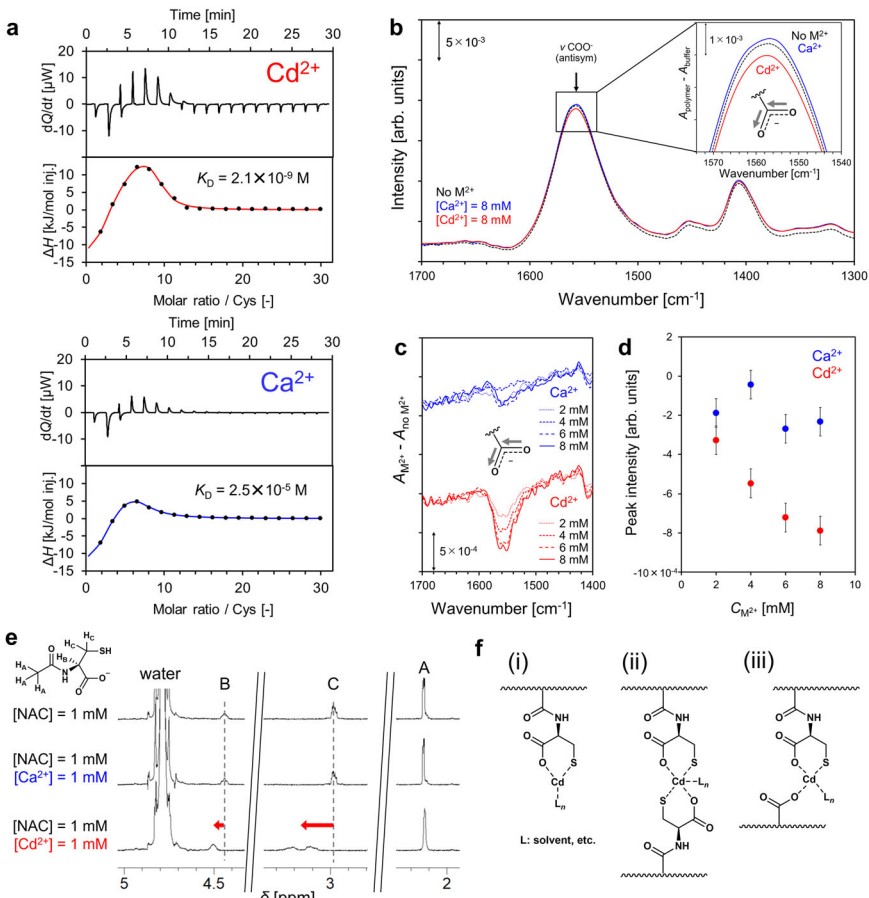

**Fig. 2 | Selective capture of Cd²⁺ ions by bio-inspired pAA-Cys5. a** Plots of additional thermal power (d$Q$/d$t$) and enthalpy (Δ$H$) versus the molar ratio of the cysteine side chain by titrating pAA–Cys5 with CdCl₂ and CaCl₂, respectively. The best-fit curves for ITC data using a two-site model are shown (solid lines). **b** ATR-IR spectra obtained by subtracting the spectrum of a buffer from that for pAA–Cys5 (4 w/v%) in the absence (black) and presence of 8 mM Ca²⁺ (blue) or Cd²⁺ (red). Inset: magnified view of the peak top for $v_{COO^-}$ (antisymmetric) band. **c** ATR-IR spectra obtained by subtracting the spectrum of pAA–Cys5 (4 w/v%) from that recorded for pAA–Cys5 (4 w/v%) in the presence of Ca²⁺ or Cd²⁺ (2, 4, 6, and 8 mM). **d** Peak intensity plotted against the concentration of Ca²⁺ or Cd²⁺ for (**c**). The data obtained in the presence of Cd²⁺ and Ca²⁺ are indicated in red and blue, respectively. Error bars corresponds to the instrumental noise determined from the baseline. **e** ¹H NMR spectra of NAC (1 mM) in the absence and presence of 1 mM of Ca²⁺ or Cd²⁺. **f** Possible structures of the Cd²⁺–pAA-Cys5 complex deduced from the NMR and FTIR spectra.

To gain further structural insights into the interaction of pAA-Cys5 with M²⁺, the attenuated total reflection-Fourier transform infrared (ATR-FTIR) spectra of pAA–Cys5 (4% w/v) were recorded in 10 mM Tris-HCl buffer (pH 7.4) in the absence and presence of M²⁺. Figure 2b presents the FTIR spectra of the polymer solutions in the absence (black) and presence of Cd²⁺/Ca²⁺ ions (red/blue). The global shape of the spectra can be characterized by two peaks at 1550 and 1400 cm⁻¹, and these peaks correspond to the antisymmetric and symmetric stretching of –COO⁻ of pAA, respectively[22]. In Fig. 2c, the differential spectra around the antisymmetric –COO⁻ band after the subtraction of the spectrum of "pAA-Cys5 with no M²⁺" are presented to highlight the changes caused by Ca²⁺/Cd²⁺ ions. The peak intensities decrease monotonically as [Cd²⁺] increase, whereas no clear trend can be detected with increasing [Ca²⁺] (Fig. 2d). Because the signals in ¹H NMR spectra of pAA-Cys5 are too broad to determine how Cd²⁺ forms a complex with –CH₂S⁻ and –COO⁻ groups precisely (Supplementary Fig. 2), we recorded the ¹H NMR spectra of a model compound, *N*-acetylcysteine (NAC), in the absence and presence of Ca²⁺ or Cd²⁺. As shown in Fig. 2e, the increase in [Ca²⁺] does not cause shifts in the positions of ¹H peaks, whereas the peaks of B and C associated with protons bound to carbon atoms adjacent to the –COO⁻ and –CH₂S⁻ groups shift in the presence of Cd²⁺, indicating that Cd²⁺ ions interact not only with –COO⁻ groups but also with –CH₂S⁻ groups. Notably, the diastereotopic splitting of CH₂ signals (C) indicated the bidentate

chelation mode[23] of NAC to Cd²⁺. Furthermore, the FTIR (Supplementary Fig. 4) and the ¹H NMR (Supplementary Fig. 5) spectra of NAC measured at different concentrations of Cd²⁺/Ca²⁺ ions provide clear evidence that Cd²⁺ ions interact with both –COO⁻ and –CH₂S⁻ groups, whereas Ca²⁺ ions do not. The chelation mechanism for phytochelatin and Cd²⁺ has been investigated using the combination of different techniques, such as UV/vis, circular dichroism, NMR, mass spectroscopy, potentiometric titration, and isothermal titration calorimetry[19,24–26]. Jalilehvand et al. combined Cd K-edge spectra and ¹¹³Cd NMR of the concentrated aqueous solution of CdCl₂ (0.5 M) and *N*-acetylcysteine (1.0 M) and proposed the multinuclear complex structures in which the position of Cd is determined by the balance of Cd−S and Cd−O bonds[14]. Wątły et al. investigated the complex stoichiometry of the well-defined oligomers, (γ-Glu-Cys)ₙ-Gly (*n* = 1 – 6), and determined the chelate stoichiometry as a function of *n*. They reported the entropic stabilization of the chelate complexes contributes to decrease the total free energy[27].

Our NMR and FTIR data on pAA−Cys5 suggest three possible structures for the Cd²⁺−pAA-Cys5 complex (Fig. 2f): (i) Cd²⁺ forms a complex with one Cys side chain, (ii) Cd²⁺ forms a complex with two Cys side chains, and (iii) Cd²⁺ forms a complex with one Cys chain and –COO⁻ group(s) of the pAA main chain. The formation of complexes between Cd²⁺ and two Cys side chains (ii) is unlikely because the separation distance between the Cys side chains (5 mol%) is non-

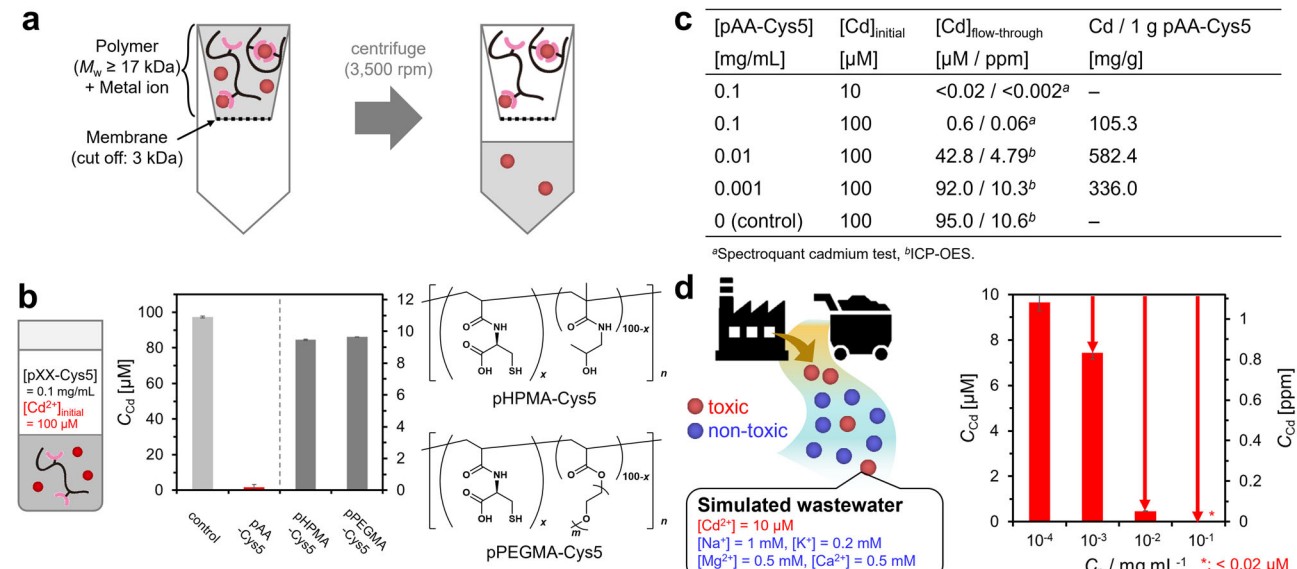

| [pAA-Cys5] | [Cd]$_{initial}$ | [Cd]$_{flow-through}$ | Cd / 1 g pAA-Cys5 |
|---|---|---|---|
| [mg/mL] | [µM] | [µM / ppm] | [mg/g] |
| 0.1 | 10 | <0.02 / <0.002$^a$ | – |
| 0.1 | 100 | 0.6 / 0.06$^a$ | 105.3 |
| 0.01 | 100 | 42.8 / 4.79$^b$ | 582.4 |
| 0.001 | 100 | 92.0 / 10.3$^b$ | 336.0 |
| 0 (control) | 100 | 95.0 / 10.6$^b$ | – |

$^a$Spectroquant cadmium test, $^b$ICP-OES.

**Fig. 3 | pAA-Cys5 is selective adsorbent for Cd$^{2+}$ ions with high loading capacity. a** Schematic for the experimental procedure. Only the ions that did not bind to pAA−Cys5 can pass through the filter because the ions bound to pAA−Cys5 polymer ($M_w \approx 1.7 \times 10^4$ Da) cannot pass the filter. **b** [Cd$^{2+}$] in the flow-through in the absence (light gray) and presence (red) of pAA−Cys5 (0.1 mg mL$^{-1}$). The corresponding data for pHPMA−Cys5 (gray) and pPEGMA−Cys5 (gray) are shown for comparison. The initial concentration of Cd$^{2+}$ ([Cd]$_{initial}$) was 100 µM. Error bars indicate the calibration accuracy of Spectroquant® colorimetric assay. **c** Table showing the initial concentrations of pAA−Cys5 and Cd$^{2+}$ ([pAA−Cys5] and [Cd]$_{initial}$, respectively) and Cd$^{2+}$ in the

flow-through ([Cd]$_{flow-through}$). These values were used to calculate the amount of Cd$^{2+}$ bound to 1 g of pAA−Cys5, whose maximum is 5.2 mmol g$^{-1}$. (**d**) Concentrations of Cd$^{2+}$ in the flow-through after mixing simulated wastewater ([Cd$^{2+}$]$_{initial}$ = 10 µM, [Na$^+$]$_{initial}$ = 1 mM, [K$^+$]$_{initial}$ = 0.2 mM, [Mg$^{2+}$]$_{initial}$ = 0.5 mM, [Ca$^{2+}$]$_{initial}$ = 0.5 mM) with $10^{-4}$, $10^{-3}$, $10^{-2}$, or $10^{-1}$ mg mL$^{-1}$ of pAA−Cys5, confirming an outstanding Cd$^{2+}$ selectivity. Error bars indicate the calibration accuracy of Spectroquant® colorimetric assay. Note that [Cd$^{2+}$] for the WHO's drinking water standard (0.03 µM) corresponds to 0.003 ppm.

uniform due to the statistical copolymerization of two monomers. As a rough estimate, if one assumes the size of AA monomer being 2.5 Å, it is plausible that the large average distance between Cys units (2.5/0.05 = 50 Å) hinders the formation of dimers. To verify if (i) is possible, two types of copolymers with the main chains containing monomers devoid of −COOH groups were synthesized, hydroxypropyl methacrylamide (HPMA, Supplementary Figs. 6 and 7) and poly(ethylene glycol methyl acrylate) (PEGMA, Supplementary Figs. 8 and 9). As presented in Supplementary Fig. 10, the ITC data for pHPMA-Cys5 and pPEGMA-Cys5 reveal no sign of interaction between the polymers and Cd$^{2+}$/Ca$^{2+}$ ions. This provides supporting evidence that Cd$^{2+}$ forms a complex with −COO$^-$ and −CH$_2$S$^-$ groups of a cysteine unit and −COO$^-$ group(s) of the pAA main chain as in (iii).

**Loading capacity and Cd$^{2+}$ selectivity of bio-inspired polymers**

The function of the bio-inspired pAA−Cys5 to treat water was tested based on two parameters: loading capacity and Cd$^{2+}$ specificity. As shown in Fig. 3a, the pAA−Cys5 solution and the buffer containing Cd$^{2+}$ were allowed to react overnight, followed by ultracentrifugation. Because the pAA−Cys5 polymer with $M_w \approx 1.7 \times 10^4$ Da cannot pass the filter with the cut-off level of $3 \times 10^3$ Da, only the ions that did not bind to pAA−Cys5 can pass through the filter. The [Cd$^{2+}$] in the ultrafiltrated flow-through was determined using inductively coupled plasma optical emission spectrometry (ICP-OES). Figure 3b presents the [Cd$^{2+}$] in the flow-through after the ultrafiltration of a mixed solution containing pAA−Cys5 (0.1 mg mL$^{-1}$) and CdCl$_2$ (100 µM), indicating the effective capture of almost all Cd$^{2+}$. In contrast, the corresponding data of the blank buffer and the two non-pAA polymers (pHPMA-Cys5 and pPEGMA-Cys5) exhibit no remarkable changes in [Cd$^{2+}$] after ultrafiltration. As the detected [Cd$^{2+}$] in the flow-through of pAA−Cys5 (<2 µM) was close to the detection limit of Cd$^{2+}$ by ICP-OES (ca. 0.1 µM), the exact value, [Cd$^{2+}$] = 0.6 µM, was determined by Spectroquant® colorimetric assay. Notably, the non-cytotoxic level of [Cd$^{2+}$][28] as well as the drinking water level defined by the World Health Organization

(WHO), [Cd$^{2+}$] ≤ 0.03 µM[29], can be achieved readily either by increasing [pAA−Cys5] or by decreasing [CdCl$_2$] in the eluent. For example, the combination of [pAA−Cys5] = 0.1 mg mL$^{-1}$ and [CdCl$_2$] = 10 µM resulted in [Cd$^{2+}$] < 0.02 µM in the flow-through.

To compare the function of pAA-Cys5 with other heavy metal ion adsorbents, we calculated the loading capacity, defined as the amount of Cd$^{2+}$ captured by 1 g of material (Fig. 3c). It should be noted here that at high polymer concentrations, the amount of bound ions per polymer becomes less with increasing polymer concentrations, because the binding sites in polymers are not saturated by Cd$^{2+}$. On the other hand, at low polymer concentrations, the equilibrium between bound and unbound states shifts towards the unbound state. Therefore, the loading capacity of pAA−Cys5 shows maximum at [pAA−Cys5] = 0.01 mg/mL based on the balance of these two effects. The maximum loading capacity of pAA−Cys5 (5.2 mmol g$^{-1}$) is 4−50 times higher than that of organic compounds, such as pAA modified with hydroquinone[30], poly(hydroxyethyl methacrylate)−cysteine conjugate[31], and poly(N-isopropyl acrylamide) hydrogel crosslinked with cysteine[32]. More remarkably, the maximum loading capacity of pAA−Cys5 is comparable to that of zeolite nanoparticles coupled to poly(vinyl alcohol) nanofibers, 7.5 mmol g$^{-1}$, which is the largest adsorption capacity to date[3,33].

The ITC data suggest that the affinity of pAA-Cys5 to Cd$^{2+}$ is by four orders of magnitude larger than the affinity to Ca$^{2+}$. To test if pAA−Cys5 can selectively capture Cd$^{2+}$, we incubated pAA−Cys5 solutions ($10^{-4}$, $10^{-3}$, $10^{-2}$, and $10^{-1}$ mg mL$^{-1}$) with a buffer containing [Cd$^{2+}$] = 0.01 mM (10 µM), [Na$^+$] = 1 mM, [K$^+$] = 0.2 mM, [Mg$^{2+}$] = 0.5 mM, and [Ca$^{2+}$] = 0.5 mM, which mimic the concentration levels in wastewater, respectively[34–36] Fig. 3d shows [Cd$^{2+}$] in flow-through, indicating that pAA−Cys5 can capture Cd$^{2+}$ even in the presence of a large excess of abundant mono- and divalent metal ions in ground water. When using 0.1 mg mL$^{-1}$ of pAA−Cys5, [Cd$^{2+}$] in flow-through was lower than the acceptable concentration declared by WHO for drinking water (<0.03 µM). These results demonstrated that

the polymer could be used to selectively remove $Cd^{2+}$ but not other ions in groundwater, such as $Na^+$, $K^+$, $Mg^{2+}$, and $Ca^{2+}$. Such a high selectivity to toxic $Cd^{2+}$ ions make them distinct from zeolites and ion exchange resins, because the ions possessing similar sizes and charges are equally captured. As presented in Supplementary Fig. 11, we also detected the decrease in $[Ca^{2+}]$ in flow-through, suggesting that $Ca^{2+}$ interact with pAA–Cys5. This is reasonable at $[Ca^{2+}] = 0.5$ mM ($5 \times 10^{-4}$ M), which is more than one order of magnitude higher than the $K_D$ value of pAA–Cys5 and $Ca^{2+}$ (~$10^{-5}$ M). As presented in Supplementary Fig. 12, the dissociation constant of pAA–Cys5 to $Mg^{2+}$ was comparable to $Ca^{2+}$ (~$10^{-5}$ M), whereas the interactions with $Na^+$ and $K^+$ could not be detected by ITC (Supplementary Fig. 12). Notably, the interaction of pAA–Cys5 and $Ca^{2+}$ could not be detected by FTIR spectra, which were measured even at higher concentrations (~$10^{-3}$ M, Fig. 2c, d). This finding can be explained by the molar fraction of Cys side chains (5 mol %). Namely, only a small portion of $-COO^-$ groups contribute to the spectral signals. In fact, the change in the spectral intensity could be detected only in differential spectra even at $[Cd^{2+}]$ ~$10^{-3}$ M, which is about six magnitudes larger than the $K_D$ value (~$10^{-9}$ M). Therefore, it seems reasonable that we could not detect the interaction of pAA–Cys5 with $Ca^{2+}$ and other cations spectroscopically.

## Hyperconfinement of bio-inspired polymers in column-based microreactor

Despite the excellent performance of pAA–Cys5, polymers in solutions cannot be directly used for water treatment. One of the most straightforward strategies to separate $Cd^{2+}$-bound pAA–Cys5 after treatment is immobilizing pAA–Cys5 chains on microparticle surfaces and packing these polymer-coated microparticles into a flow-through reactor. Supplementary Fig. 13 shows the synthetic scheme for the end-functionalized pAA–Cys5 prepared to coat the surfaces. To get the first proof of principle, we coated silica microparticles (diameter: 10 μm) with bilayer lipid membranes incorporating 2 mol% of biotinylated lipids, coupled pAA–Cys5–biotin (Supplementary Figs. 14–19) to the membrane via neutravidin crosslinker (Fig. 4a, Supplementary Fig. 20), and load them in a fast protein liquid chromatography (FPLC) column (inner diameter: 10 mm)[37]. Note that the grafting of pAA–Cys5 brushes on silica particles have two advantages. First, as demonstrated previously[38–41], the average distance between polymer chains $\langle d \rangle$ can

be controlled by the molar fraction χ at a nm-accuracy,

$$\langle d \rangle = \sqrt{A_{lipid}/\chi} \quad (1)$$

where $A_{lipid}$ is the cross-sectional area of one lipid molecules, ($A_{lipid} \approx 0.6$ nm²)[42]. For example, the inter-polymer distance corresponding to $\chi = 0.02$ can be estimated as $\langle d \rangle \approx 5.5$ nm. Second, the grafting of brushes on silica particles makes the separation of the polymer-$Cd^{2+}$ complex much simpler. The use of pAA–Cys5 gels / particles is not practically possible, because the density of the hydrated polymers is very close to the density of the medium. The homogeneous coating of the particle surface with pAA–Cys5 was confirmed by confocal fluorescence microscopy, using fluorescently labeled polymer (Fluor–pAA–Cys5–biotin, Supplementary Fig. 21). The scanning electron microscopy image of the particle surface and the three-dimensional image of pAA–Cys5-coated microparticles reconstructed from the confocal microscopy images are presented in Supplementary Fig. 22. The functionalized particles (0.85 g in weight) were packed in a compact, flow-through microreactor, which enabled the confinement of silica microparticles (surface area of 1.1 m²) functionalized with 0.6 μmol of pAA–Cys5 in a volume of 1.8 mL[37]. A buffer (10 mM Tris-HCl, pH 7.4) containing 100 μM CdCl₂ was fed by a syringe pump at the flow rate of 0.02 mL min⁻¹, led through the column, and fractionated (Fig. 4b).

$[Cd^{2+}]$ of each fraction was determined by conducting colorimetric assays, and the data were plotted as a function of the fraction number (Fig. 4c). As the volume of each fraction is 1 mL, the term "fraction number 10" indicates a "total eluent volume of 10 mL." Notably, $[Cd^{2+}]$ of fractions 1–36 remained below the detection limit of the assay, which fulfills the WHO's drinking water level. These data indicate that this prototype can be used to treat 36 mL of a concentrated solution of CdCl₂ (100 μM) to bring down the concentration of the contaminant to a level that meets the appropriate standards for drinking water (<0.03 μM). The total amount of $Cd^{2+}$ captured by the microreactor was calculated based on the cumulative summation of $[Cd^{2+}]$ of each fraction (Fig. 4d). Here, $8.2 \times 10^{-4}$ g, corresponding to $4.9 \times 10^{-8}$ mol, of polymers were grafted to 0.85 g of silica microparticles. This yields the estimated maximum loading capacity of $Cd^{2+}$ per 1 g of polymer to be 11 mmol g⁻¹. Taking the number-average molecular weight ($M_n$) of pAA–Cys5 ($8.0 \times 10^3$) and pAA–Cys5–biotin

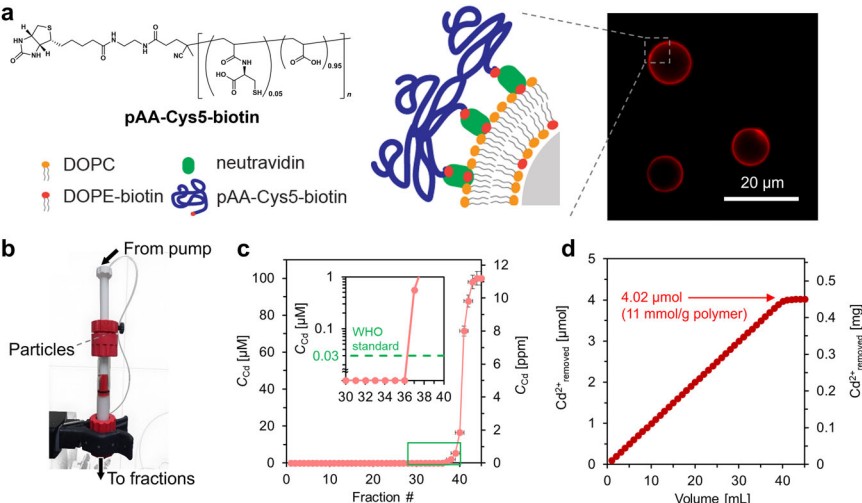

**Fig. 4 | Hyperconfinement of pAA-Cys5 into column-based microreactor prototype. a** Functionalization of silica microparticles (diameter: 10 μm). The homogeneity of the surface coating was confirmed by the confocal fluorescence image of silica microparticles coated with fluorescently labeled pAA–Cys5–biotin (Fluor–pAA–Cys5–biotin, Supplementary Fig. 21). **b** The flow-through microreactor prototype based on an FPLC column that enables the confinement of silica

microparticles (surface area of 1.1 m²) functionalized with 0.6 μmol of pAA–Cys5 in a volume of 1.8 mL. **c** $[Cd^{2+}]$ of the eluent plotted as a function of the fraction number. Inset: magnified plots near the onset. Error bars for x- and y-axes indicate the fraction volume and the calibration accuracy of Spectroquant® colorimetric assay, respectively. **d** Total amount of removed $Cd^{2+}$ plotted as a function of elution volume. $[Cd^{2+}]$ satisfied the concentration criteria recommended by WHO for drinking water.

$(7.4 \times 10^3)$, the difference in the maximum loading capacity of polymers with and without confinements are estimated to be 41.6 mol mol$^{-1}$ and 79.8 mol mol$^{-1}$, respectively. Currently, we interpret the higher loading capacity realized by the particle-grafted pAA−Cys5−biotin in terms of the neighbor effect or the multivalent binding[43–45]. The former is a well-established concept in understanding the effect of adjacent ligands on the binding to a linear lattice. The latter is the principle of a targeted strengthening of interactions between the binding partners forming cooperative, multiple interactions that are based on individually weak, noncovalent bonds. In both cases, the presence of binding partners in the vicinity strengthens the overall binding. As suggested by the NMR analysis, the determination of valency of interaction, i.e. the number of contributing moieties in the chelate formation, is difficult in our experimental systems. Nevertheless, it is plausible that Cd$^{2+}$ that escapes from one complex can readily form another complex by interacting with a neighboring −CH$_2$S$^-$ and −COO$^-$ pairs. Yet, using this microreactor prototype, it takes about 24 h to treat 36 mL of water, which is too slow for the water treatment in practice. This extremely low flow rate is caused by a high back pressure $P$, which scales with:

$$P \propto \frac{L \times F}{D^2 \times ID^2} \qquad (2)$$

$L$ is the length of the path (packed beads), $F$ the flux, $D$ the particle diameter and $ID$ is the inner diameter of the column. The use of larger particles decreases the pressure if $L$ is kept constant, e.g. the use of 20 μm-large particles reduces the $P$ by a factor of 4. However, in order to keep the same surface area, one needs 2 times larger bed volume and hence the length $L$. Therefore, we concluded that the prototype was useful to demonstrate the proof of principle but not suited for achieving a realistic flow rate comparable to a fixed-bed column system[46].

## Integrative water treatment systems for high-throughput, selective removal of Cd$^{2+}$ ions

Next, we tried to increase the efficiency of our flow-through reactor towards the realistic application in water treatment by increasing the area-to-volume ratio and the flow rate. The area-to-volume ratio can be increased using small particles. For example, the decrease in the particle diameter by a factor of 8 results in the increase in the area-to-volume ratio by a factor or 512/64 = 8. However, as described above, the use of small particles results in a significant increase in the back pressure $P$, because

$$P \propto D^{-2} \qquad (3)$$

Therefore, to realize both a higher area-to-volume ratio and a lower back pressure, we designed a new reactor. Small silica microparticles (diameter: 1.2 μm) were used to achieve a high area-to-volume ratio, and the surface of microparticles was functionalized with pAA−Cys5 coupled to the head group of dioleoylphosphatidylethanolamine (DOPE), called pAA−Cys5−DOPE (Fig. 5a and Supplementary Figs. 23-25). The deposition of a lipid monolayer incorporating pAA−Cys5−DOPE (2 mol%) on the surface of silanized silica microparticles[47] enables the one-step immobilization of pAA−Cys5 onto microparticles. We changed the coating protocol for smaller particles (diameter 1.2 μm) to minimize the risk of particle aggregation. Once the beads form aggregates, they could hardly be resuspended as single particles. The surface functionalization with a lipid monolayer incorporating pAA−Cys5−DOPE (2 mol%) following our previous account[47] prevented the undesired aggregation of particles caused by the multiple centrifugation cycles (Supplementary Fig. 20). The covalent coupling of a pAA−Cys5 chain directly to the lipid head group also ensure the stability of the polymer chains, which was

another positive effect. The molecular sieve (pore diameter: 0.1 μm) used in the first prototype (Fig. 4) was replaced by a cellulose membrane coated with chitosan functionalized with N-succinimidyl (NHS)-terminated pAA−Cys5 (chitosan−g−pAA−Cys5, Fig. 5b) to reduce the back pressure and hence to increase the flow rate. The synthesis of chitosan−g−pAA−Cys5 is presented in Supplementary Figs. 26 and 27.

The water treatment capacity was tested by packing a cellulose/chitosan−pAA−Cys5 membrane (Supplementary Fig. 28) and small silica microparticles (1.2 μm) coated with pAA−Cys5−DOPE into an Amicon® chamber (inner diameter: 45 mm) (Fig. 5c). Here, 0.063 g, corresponding to $8.9 \times 10^{-6}$ mol, of pAA−Cys5−DOPE deposited on 5 g of silica microparticles yielding the polymer/bead mass ratio of 0.013. We monitored how this system can treat the water containing 10 μM CdCl$_2$, which replicated the concentration level in industrial wastewater. Figure 5d shows the [Cd$^{2+}$] in the eluent plotted as a function of the total elution volume $V$. The open symbols are the concentrations determined only in the presence of the cellulose/chitosan−pAA−Cys5 membrane, whereas the solid symbols correspond to the data obtained in the presence of both the cellulose/chitosan−pAA−Cys5 membrane and the polymer-functionalized microparticles. Notably, the new system could be operated at a flow rate of 5 mL min$^{-1}$, which is 250 times higher than the flow rate used for the previously described prototype (Fig. 4). Even though the diameter of the microparticles (1.2 μm) was almost by one order of magnitude smaller than those used for the previous prototype (10 μm), the back pressure remained <1 bar, which is well below the operational limit specified for a commercial chamber (3.8 bar). The flow rate of our system (300 mL h$^{-1}$) is comparable to those of fixed-bed column systems, ~10-1000 mL h$^{-1}$ [46]. The linear velocity, a system size-independent index of water permeability, of our system was 0.2 m h$^{-1}$, which is also comparable to those of conventional systems, ~0.1-1 m h$^{-1}$ [46].

In the presence of unmodified cellulose membrane and silica microparticles without polymers, the system could not remove Cd$^{2+}$, showing a significant leakage from the very first fraction (Fig. 5d). Even in the absence of functionalized microparticles, the cellulose/chitosan−g−pAA−Cys5 membrane could remove [Cd$^{2+}$], but a distinct leaking of Cd$^{2+}$ could be detected already at $V = 60$ mL (open symbols, Fig. 5d). The loading of different amounts of chitosan−g−pAA−Cys5 exhibited a nonlinear dependence of the onset volume of the leakage on the amount of grafted chitosan−g−pAA−Cys5 (Supplementary Fig. 29). This suggests that the combination of the cellulose/chitosan−g−pAA−Cys5 membrane and the pAA−Cys5-coated particles is necessary, because the membrane alone is not sufficient to capture Cd$^{2+}$ under the practical flow rate used for the water treatment (5 mL min$^{-1}$). In contrast, in the presence of both the cellulose/chitosan−pAA−Cys5 membrane and the pAA−Cys5-coated microparticles, the [Cd$^{2+}$] level remained below the detection limit of the colorimetric assay up to 300 mL at a flow rate of 5 mL min$^{-1}$ (solid symbols, Fig. 5d). Therefore, the obtained data indicate that the combination of the cellulose/chitosan−pAA−Cys5 membrane and pAA−Cys5-functionalized microparticles (5 g) packed in ~3 mL volume can be used to treat 0.3 L of industrial wastewater within 1 h. The Cd$^{2+}$ removal capacity of our system was further examined by treating the simulated wastewater containing [Cd$^{2+}$] = 0.01 mM (10 μM), [Na$^+$] = 1 mM, [K$^+$] = 0.2 mM, [Mg$^{2+}$] = 0.5 mM, and [Ca$^{2+}$] = 0.5 mM. The [Cd$^{2+}$] level remained below the detection limit of the colorimetric assay up to 135 mL at a flow rate of 5 mL min$^{-1}$ (Supplementary Fig. 30).

The regeneration and recovery of the materials can potentially be achieved by the following two strategies. In the first strategy, the polymers can be regenerated by cancelling the chelate complex by adding a competitor, such as ethylenediaminetetraacetic acid (EDTA). As presented in Supplementary Fig. 31, the continuous treatment with the buffer containing 10 mM EDTA for 200 min resulted in the recovery of function by 83%. Although this suggests the recyclability of the materials, this treatment is to simply concentrate Cd$^{2+}$ ions from one

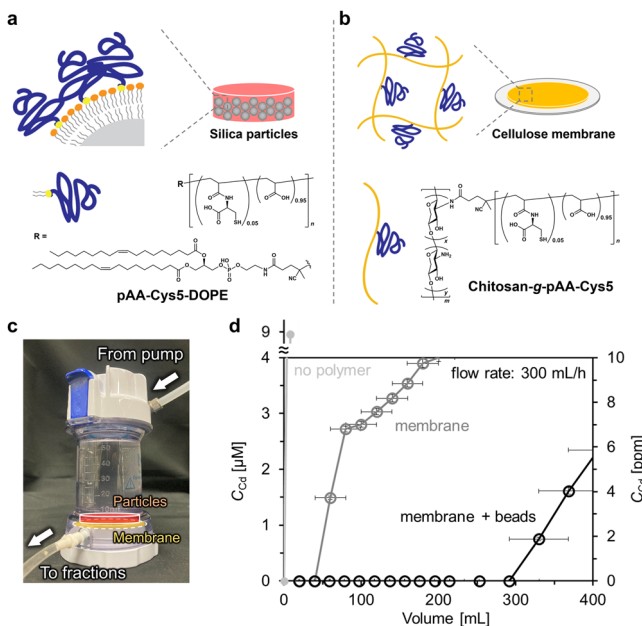

**Fig. 5 | High-throughput, integrative water treatment system based on pAA−Cys5−functionalized membrane and particles.** Combination of (**a**) silica microparticles (diameter: 1.2 μm) functionalized with pAA−Cys5 and (**b**) cellulose membrane coated with chitosan functionalized with pAA−Cys5. The surface of silanized silica microparticles was functionalized with a lipid monolayer incorporating 2 mol% of pAA−Cys5−DOPE. **c** Photograph of the device consisting of the microparticles and a membrane functionalized with pAA−Cys5. **d** [Cd²⁺] in the eluent plotted as a function of elution volume. The unmodified cellulose membrane and silica microparticles without polymers showed a significant leakage from the very first fraction (light gray). The cellulose/chitosan−*g*−pAA−Cys5 membrane showed a leakage of Cd²⁺ already at $V = 60$ mL (gray). In contrast, the combination of the membrane and the pAA−Cys5-functionalized microparticles (black) could remove 10 μM Cd²⁺ (typical industrial wastewater level) from 0.3 L in 1 h down to the level approved for drinking water. Error bars for *x*- and *y*-axes indicate the fraction volume and the calibration accuracy of Spectroquant® colorimetric assay, respectively.

aqueous phase (polluted water model) to another (EDTA buffer). In the second strategy, the heavy metal and silica particles can be recovered by taking the difference in temperature conditions for decomposition into account. For example, the organic compounds can be removed and the metal (Cd) can be recovered in a fluid phase at 600 °C, which is well below the melting temperature of silica (1710 °C). Although the potential recovery of heavy metal can add a unique advantage to our materials over zeolites and other porous inorganic materials, this approach might not be helpful for the minimization of global carbon footprints.

The positive proof-of-concept obtained in this study suggests that the spatial confinement of bio-inspired, synthetic polymer materials realizes potentially recyclable flow-through systems for the high-throughput, selective removal of harmful substances from environmental water, exploiting specific recognition functions of biological systems.

## Discussion

We designed and fabricated a flow-through water treatment system in an attempt to provide an alternative to phytoremediation, which relies on tree planting. The system is based on a copolymer material that draws inspiration from highly conserved phytochelatin protein in plants. The copolymer (pAA−Cys5) consists of AA monomers with cysteine side chains (5 mol%). The ITC data indicate that the dissociation constant of pAA−Cys5 to Cd²⁺ ($K_D = 2.1 \times 10^{-9}$ M per molecule) is four-to-five orders of magnitude lower than those reported for

peptides mimicking the sequence of endogenous phytochelatins. Moreover, the $K_D$ value of pAA−Cys5 to Ca²⁺ and Mg²⁺ is four orders of magnitude higher than that recorded for Cd²⁺. Furthermore, the interactions with Na⁺ and K⁺ could not be detected by ITC, suggesting that pAA−Cys5 can selectively capture Cd²⁺. Intriguingly, the dissociation constant of pAA−Cys5 to Hg²⁺ ($K_D = 7.1 \times 10^{-9}$ M per molecule) is in the same order of magnitude as that to Cd²⁺ (Supplementary Fig. 32). As Hg²⁺ was reported to be captured by plant phytochelatin[48], this result suggests that pAA−Cys5 is able to capture Hg²⁺, as plant phytochelatin does. The FTIR and NMR data suggest that pAA−Cys5 formed a complex with Cd²⁺ but not with Ca²⁺, which explains the differential affinities of pAA−Cys5 for Cd²⁺ and Ca²⁺. Notably, the maximum loading capacity of (5.2 mmol g⁻¹) of pAA−Cys5 determined by the batch test, is comparable to the highest value reported so far (7.5 mmol g⁻¹).

To verify the applicability of the system for water treatment, the spatial hyperconfinement of pAA−Cys5 was attempted by grafting the copolymers onto silica microparticles (10 μm) using biotin−neutravidin crosslinkers. pAA−Cys5 (0.6 μmol) was grafted on the total surface area of 1.1 m², and the polymer-functionalized microparticles were hyperconfined in an HPLC column with a bed volume of 1.8 mL. Although this microreactor prototype could remove Cd²⁺ from 36 mL of polluted water containing 100 μM Cd²⁺, it is operative only at a very low flow rate (0.02 mL min⁻¹) because of the high back pressure. To increase the area-to-volume ratio, smaller silica microparticles (1.2 μm) were functionalized by the deposition of a lipid monolayer incorporating pAA−Cys5 coupled to a lipid head group (pAA−Cys5−DOPE). To avoid the high back pressure, the pAA−Cys5-functionalized silica microparticles were supported by a cellulose membrane coated with chitosan functionalized with pAA−Cys5. The direct deposition of a lipid monolayer incorporating pAA−Cys5−DOPE promotes an increase in the area-to-volume ratio and prevents particle aggregation, whereas the combination of the functionalized particles and the pAA−Cys5-coated membranes helps reduce the back pressure in the system. This resulted in an increase in the flow rate by a factor of 250, which enabled reduce the Cd²⁺ concentration in 0.3 L water from the typical industrial wastewater level (10 μM) down to the level recommended for drinking water (<0.03 μM) within 1 h. The results suggested that the spatial hyperconfinement of bio-inspired, synthetic polymer material integrated into flow-through systems is a promising alternative to phytoremediation, which realizes the highly efficient and selective removal of heavy metal ions from aquatic environments.

## Methods
### Materials

*S*-Trityl-ʟ-cysteine, 2-(dodecylthiocarbonothioylthio)-2-methylpropionic acid (DDMAT) (>98%), poly(ethylene glycol) methyl ether acrylate (PEGMA), and chitosan (medium molecular weight) were purchased from Sigma-Aldrich (St. Louis, MO, USA). Potassium hydroxide (KOH) (>85%), sodium hydroxide (NaOH) (>97%), acryloyl chloride (>98%), acrylic acid (AA) (>98%), dimethyl sulfoxide (DMSO) (>99%), acetone (>99.5%), hexane (>96%), trifluoroacetic acid (TFA) (>98%), diethyl ether (>99.5%), 4,4′-azobis(4-cyanovaleric acid) (ACVA) (>98%), *N*-hydroxysuccinimide (NHS) (>98%), *N*,*N*-dimethylformamide (DMF) (>99.5%), molecular sieves 4 A, (+)-biotin (>97%), *N*,*N*′-carbonyldiimidazole (CDI) (>95%), ethylenediamine (>99%), dichloromethane (DCM) (>99.5%), triethylamine (Et₃N) (>99%), 1,2-dioleoyl-sn-glycero-3-phosphoethanolamine (DOPE) (>95%), and *N*-acetylcysteine (NAC) (>98%) were obtained from FUJIFILM Wako Pure Chemical Co. (Osaka, Japan). *N*-(2-hydroxypropyl)methacrylamide (HPMA) (>98%) was purchased from Tokyo Chemical Industry Co. (Tokyo, Japan). 1-(3-Dimethylaminopropyl)-3-ethylcarbodiimide hydrochloride (EDC·HCl) was purchased from Peptide Institute (Osaka, Japan). Azobisitobutyronitrile (AIBN) (>99.5%) was purchased from Kishida Chemical Co. Ltd. (Osaka, Japan).

## Measurements

$^1$H Nuclear Magnetic Resonance (NMR) spectra were recorded at 400 MHz using a JNM–ECS400 NMR spectrometer (JEOL, Tokyo, Japan). The chemical shifts were referenced with respect to those of the NMR solvents ($\delta$ = 2.49 and 7.26 ppm for DMSO-$d_6$ and CDCl$_3$, respectively). Silica gel column chromatography was performed using a Biotage Isolera One (Biotage AB, Uppsala, Sweden) equipped with a SNAP Ultra Column cartridge. Gel permeation chromatography (GPC) of the polymers was carried out using an HPLC system (CBM-20A/LC-20AD/SIL-10AXL/DGU-20A3R/CTO-20AC, Shimadzu, Kyoto, Japan) equipped with an SB-804 HQ column (Shodex, Tokyo, Japan) and a refractive index (RI) detector (RID-20A, Shimadzu, Kyoto, Japan). The experiments were conducted at 25 °C using Tris-HCl buffer (10 mM) containing NaCl (100 mM) as an eluent at a flow rate of 0.7 mL min$^{-1}$. ReadyCal-Kit PEG (PSS Polymer Standards Service GmbH, Mainz, Germany) was used as the calibration standard. ITC data were recorded using a MicroCal PEAQ-ITC instrument (Malvern Panalytical, Malvern, UK). ICP-OES spectra were recorded using an SPS7800 instrument (Hitachi High-Tech, Tokyo, Japan). The UV-visible absorption spectra were recorded using a Shimadzu UV-2600 spectrometer (Shimadzu, Kyoto, Japan). Dynamic light scattering (DLS) profiles were recorded using an ELSZ-2000ZS instrument (Otsuka Electronics, Osaka, Japan). Fourier-transformed infrared (IR) spectral profiles of the polymers were recorded at the SPring-8 BL43IR beamline using a Bruker Vertex70 spectrometer (Bruker, Billerica, US) equipped with an ATR system (MicromATR Vision, Czitek, Danbury, US) attached using a diamond crystal. The wavenumber resolution was 2 cm$^{-1}$. The IR spectral profiles of NAC were recorded using a JASCO FTIR-4100 spectrometer (JASCO, Tokyo, Japan) equipped with an ATR system attached using a ZnSe crystal with the wavenumber resolution 4 cm$^{-1}$.

## Preparation of cysteine-containing polymers

Vinyl polymers carrying cysteine residues and various comonomers (pXX-Cys5) were synthesized through reversible addition–fragmentation chain-transfer (RAFT) radical polymerization of S-Tri-Cys-AAm with a comonomer (AA, PEGMA, or HPMA). AIBN was used as an initiator, and DDMAT was the chain transfer agent. The process was followed by the deprotection of the trityl group with TFA. Briefly, S-Tri-Cys-AAm, the comonomer, AIBN, and DDMAT were dissolved in DMSO which was dried using molecular sieves (4 A). The solution was purged with nitrogen gas for 1 h, sealed, and heated overnight in an oil bath thermostated at 65 °C. The solution was then poured into 10-fold volume of stirred acetone. The resulting viscous concentrated phase was collected following the process of centrifugation. The supernatant was decanted, following which TFA was added to the residue, and the mixture was stirred overnight. The resulting solution was poured into 10-fold volume of diethyl ether in the cases of AA and HPMA or a mixed solvent system containing diethyl ether and hexane (1/1, v/v) in the case of PEGMA. The resultant precipitate was washed three times with the same amounts of diethyl ether in the cases of AA and HPMA or a mixed solvent of diethyl ether and hexane (1/1, v/v) in the case of PEGMA. The remaining solid samples were dried under reduced pressure at room temperature (ca. 25 °C). Detailed experimental procedures are provided in the Supplementary Information.

## Data availability

The data that support the findings of this study are available within the article and its Supplementary Information file, and from the corresponding authors upon request.

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

## Acknowledgements

The authors are grateful to Prof. H. Sato (Kobe University) for insightful comments on the IR analysis. The authors thank Prof. T. Hirai and Prof. Y. Shiraishi (Osaka University) for their help in the ICP measurements, Dr. P. Linke, Z. Hajian Foroushani, and T. Häßner (Heidelberg University) for assisting with the column experiments and the SEM imaging, and Dr. U. Engel (Nikon Imaging Center, Heidelberg) for supporting the confocal microscopy imaging. M.N. thanks Prof. A. Hashidzume (Osaka University) for fruitful discussions and comments. This study was supported by JSPS KAKENHI (JP19H05714 to M.N., Y.I. and T.N., JP19H05717 to Y.I., JP20H05202 and JP22H04519 to T.N., JP19H05719 to M.N. and M.T.) and the German Science Foundation (SPP2171 Ta253/14–2 and Germany's Excellence Strategy—2082/1—390761711 to M.T.). M.T. thanks the Nakatani Foundation for its support. A.Y. thanks the L-INSIGHT Program of the Ministry of Education, Culture, Sports, Science and Technology (MEXT), Japan. The authors thank Editage for editing the early version of the draft.

## Author contributions

Conceptualization and supervision: M.N., M.T.; synthesis and characterization: A.S., M.N.; IR measurements and analysis: Y.I., M.N.; NMR measurements and analysis: T.N., M.N.; imaging, column experiments and analysis: A.D., J.S., A.S., A.Y., M.N., S.S., S.K., M.T.; writing–original draft: M.N., M.T.; reviewing and editing: Y.I., T.N., S.S., S.K. All authors proofread, commented on, and approved the final version of this manuscript.

## Funding

## Competing interests

The authors declare no competing interests.
