## [Peer Review File · Nature Communications]

REVIEWER COMMENTS

Reviewer #1 (Remarks to the Author):

In the present work, the authors test polyacrylic acid-based copolymer that they say is inspired by plants. They then examined the ability of the polymer to remove Cd²⁺ from water under continuous flow after coating silica beads and a membrane surface with the polymer. The prepared polymer consists of poly acrylic acids functionalized with cysteine side chains. Notably, they previously reported the same polymer (<https://pubs.rsc.org/en/content/articlelanding/2022/na/d2na00350c>), deposited it on a membrane, showed significant difference in response to Cd over Ca, and showed that the Cd interactions stems from both COOH and SH functionality. Given this, I believe the novelty of the current work stems from demonstrating the materials performance for Cd removal from water, the proposed mechanism for chelation, and the strong affinity for Cd²⁺ over Ca²⁺ unveiled by ITC measurements. In particular, given the high affinity and that mechanistic insight is often missing in the literature, I may be interested in seeing this published in the current journal; however, there are some comments the authors should address below that can help them improve the manuscript. In particular, they should test the material in a competitive environment containing Cd and Ca and many other competing ions commonly found in water sources (and in relevant concentrations) under continuous flow and also provide insight into whether the Cd can be released, and provide all detailed experimental procedures.

1. The authors say that their polymer was inspired by phytochelatin. While it is understood that the two have similar functionality, it is not clear that that functionality is arranged similarly according to figure 1 and so one would expect that the function of the two distinct polymers to be different. Can the authors comment? For instance, is the chelation mechanism for phytochelatin and Cd reported? What are the differences in the binding strength of Cd?

2. The authors should report precise procedures for everything. This includes what specifically they did for the preparation of the polymer, including the cysteine grafting, the coating on the silica beads and membrane preparation. This should include fine details like quantities of starting materials, where starting materials came from, temperatures, solvents, reaction times, etc, so that the work can be reproduced. They should also be sure all details of all experiments used to test the metal ion removal is listed, how materials were regenerated, and tested under continuous flow.

3. The authors state, "By grafting of bio-inspired copolymers onto silica microparticles and cellulose membranes, 0.1 g (~ 10¹⁷ molecules) copolymers can be confined in ~ 3 mL volume. Its not clear what the importance of this is, nor what they are referring to when they say 10¹⁷ molecules. Are they trying to state the number of monomers in 0.1 gram of polymer? It's an awkward way to put things and I am not sure is very relevant.

4. To improve the readers ability to follow what they are doing can they also provide the concentration of the metal ions in ppm and ppb rather than molarity or at least both?

5. They talk about the polymers performance first, but it's a little misleading as the polymers must be immobilized onto other particles for water purification applications. So, to look at the performance of the polymer by mass is misleading in my opinion. So, maybe if they state specifically why they did this study upfront with only the polymer, albeit not possible to use in this format for water purification studies, it would be helpful. The reason I say this is because at the moment, the section seems rather irrelevant.

6. Its recommended that the authors include more controls in their study. They should consider looking at the performance of the non-functionalized PAA, PAA-silica, and PAA-cellulose membrane, and just the cellulose membrane, along with the ones they have included (PAA-Cy5, the PAA-Cy-5-silica, and PAA-Cy5-cellulose membrane).

7. The authors state: "As shown in Fig. 3a, the pAA-Cys5 solution and the buffer containing Cd²⁺ were allowed to react overnight, and the [Cd²⁺] in the ultrafiltrated flow-through was determined using inductively coupled plasma optical emission spectrometry (ICP-OES)" Its really unclear to me what they mean by ultrafiltrated flow through....are they intending to say that they are using ultrafiltration to remove the cadmium from the solution that contains suspended polymers? If they give a more precise description here it would be helpful as what they are talking about as this is not commonly encountered in water purification studies. I had to read this section multiple times and stare at the image a bit to really follow what they are doing.

8. Along the same lines above, Figure 3 a is not clear. They need to provide some kind of labels to guide the reader in what they are looking at. After looking at it a bit, I finally think I understand, but it should not really be this difficult.

9. What is the molecular weight of the polymer they are producing? Why is it that they cannot make polymeric beads, as found in resins, rather than putting it onto a silica support or onto a membrane?

10. The authors change their experiment when they look at pAA-Cys5 in the presence of cadmium only and then with cadmium and calcium, they change their experiment. It would be good if they can do it the same way (0.01mg/mL pAA-Cys5) and then 10mM Cd²⁺ and 500mM Ca²⁺. Then figure

3b and 3d could be combined. Also, where are the error bars on 3d? They should be sure to provide error bars (and esds in the text) when possible, for all experiments.

11. Along the same lines of comment 10, they are adsorbing a lot of Ca^{2+} because the concentration of Ca^{2+} is much higher than Cd. Thus, I think they have as much Ca or even more than Cd in the polymer, despite the higher affinity towards Cd^{2+} . For example, they remove almost 50% of the Ca^{2+} concentration at a polymer loading of 0.1 mg/mL. If they see no change in the IR spectrum associated with the polymer in the presence of Ca^{2+} , then what do they suppose is the nature of the interaction between Ca and their polymer? Given this, I would like to see how the Ca and other ions impact the materials performance in the final flow through experiment, like Na, K, Ca, among others.

12. In the experiments shown in Table 3C. Why is it that when they decrease the pAA-Cys5 loading to 0.001 they see a decrease in capacity? This does not make any sense to me. Should it not be as high as that observed at 0.01 or higher? If there are esds associated with these measurements.

13. The authors state, "The estimated maximum loading capacity per 1 g of polymer was 11 mmol g^{-1} ." It should instead be "1 gram of polymer is 11 mmol". Also, considering that the polymer cannot be employed here as the active material alone and they need a support, they should also report the total mass of the coated beads that would be required to remove 11 mmol of Cadmium. Otherwise, it is deceiving.

14. I think they should get rid of these small cartoons they are using in figure 3d unless they modify it to make it clearer to the reader what they are looking at.

15. The authors state "The obtained capacity is two times higher than that recorded in the solution system (5.2 mmol g^{-1}), which can be attributed to the fact that the polymers were confined within a small volume." Its not clear why the capacity of the material on a per gram basis would increase with respect to the volume. Its very awkwardly written. While the sentence after it clarifies, they should rewrite this to be more concise.

16. Can the authors comment on why it takes about 24 h to treat 36 mL of water? Why are they using such low flow rates, what are the pressures? Could this not be overcome by using larger silica beads rather than the 10 micron ones. I think you can buy commercial silica beads with much larger diameters? Or is there some other reason that the flow was so low? More explanations could help readability.

17. The section on increasing the efficiency of their flow through setup is confusing. They keep saying they are increasing the area to volume ratio but never discuss what they are increasing the area to volume ratio of. I am assuming they are simply trying to increase the quantity of polymer by using smaller silica particles which offer a larger surface area to volume ratios. They need to revise this section as its very hard for readers to follow.

18. Why did they change the coating process when they reduced the size of the silica beads?

19. The authors state, "The molecular sieve (pore diameter: 0.1 μm) used in the first prototype (Fig. 4) was replaced by a cellulose membrane coated with chitosan functionalized with N-succinimidyl (NHS)-terminated pAA-Cys5 (chitosan-g-pAA-Cys5, Fig. 5b) to achieve high flux under conditions of low back pressure. First, I am assuming the membrane they use somehow provides a much better flow rate over the molecular sieve. If that is the case, they should clearly state this. Second, why do the authors prepare a membrane coated with pAA-Cys5 as well as the coated beads in the same separation unit? I am assuming the membrane is just there to inhibit the loss of their beads. They should also do a control here to show that the Cd goes through the non-functionalized membrane. I did not see this, unless I missed it, and do the experiment with only the functionalized membrane as well.

20. Notably, 5 grams of material (beads and polymer, I presume) to purify 0.3 L of water is a lot, which may be further optimized later. So, they should also describe the overall mass of polymer used in this flow through setup. What is the polymer/bead mass ratio in this case?

21. The state that that the polymer can be burned off to recover the cadmium. To state this, they should give protocols, unless I missed it for how the cadmium is retrieved and then the methods to clean the silica surface and recoat the beads and report the performance after bead reuse.

22. I would also be interested to know if they can regenerate the polymer using other chemical means, or is the Cd so strongly bound that one must pyrolyze the material?

23. The authors should work on the overall paper flow. It was difficult for me to read and hence I had to go over it several times.

Reviewer #2 (Remarks to the Author):

In this work, phytochelatin-inspired copolymers containing carboxylate and thiolate moieties were synthesized, and grafted onto silica nanoparticles and cellulose membranes for the removal of Cd ion from contaminated water in a flow-through system. The copolymers consisting of poly(acrylic acid) partially functionalized with cysteine side chains were synthesized via reversible addition-fragmentation chain-transfer (RAFT) radical copolymerization of acrylic acid (AA) and AA coupled to cysteine, and was referred to as pAA-Cys5. It was shown that the cadmium removal effectiveness (representing by the dissociation constant for Cd ion) of pAA-Cys5 is four-to-five orders of magnitude higher than that for peptides mimicking the sequence of endogeneous phytochelatin. Isothermal titration calorimetry, FTIR, and NMR spectroscopy were used to elucidate the structures of the polymer-ion complexes.

In general, the work represents important efforts in selective removal of cadmium from contaminated water to provide drinking-water with acceptable traces of cadmium. Control experiments and characterization were carefully conducted to elucidate the main complexation mechanism between Cd ions and pAA-Cys5. For example, two other copolymers with the main chains containing monomers devoid of carboxyl groups (i.e., pHPMA-Cys5 and pPEGMA-Cys5) were tested to access the role of carboxyl groups on PAA in complexation with Cd ion. It was hypothesized and empirically confirmed that cadmium ion forms a complex with one Cys chain and carboxyl group(s) of the PAA main chain.

While interesting insight is provided, from the reviewer's perspective, fundamental insights and engineering breakthrough gathered from this work are not at the level of novelty suitable for publishing on Nat. Commun. Fundamentally, while interesting, this work does not address the main challenges or scientific gaps encountered in this field of separation, which involve 1) the separation between similar ions, or 2) the separation of a target ion, such as Cd ion, from a complex multi-ion mixture. While the authors claim that the proposed material system offers "specific recognition functions of biological systems" (page 16), the ion specificity/recognition capability of pAA-Cys5 toward Cd ion has not been well-understood nor demonstrated in-depth. Merely testing the separating performance of pAA-Cys5 in a mixture of two divalent cations with different ion properties (e.g., Cd and Ca ions) does not sufficiently demonstrate the ion recognition and discrimination capability of the material. Even among the separation of these two divalent cations, it is unclear why pAA-Cys5 has a stronger affinity toward Cd ion than Ca ion. While the authors successfully elucidated the mechanism at which Cd ion form a complex with the pAA-Cys5 system, the main mechanism at which the two divalent cations are discriminated by pAA-Cys5 is not discussed. In-depth characterization and discussion are essential to provide more insight governing the behaviors of pAA-Cys5.

In addition, more work is needed to demonstrate a reliable and robust engineering system to evaluate the separating performance of pAA-Cys5 either in the particle or membrane platform. Current results and discussion have not been well-developed because they lack of in-depth analyses and discussion beyond a random engineering demonstration. Data are shown without error bars, and filled with lots of cartoon demonstrations. While these cartoon demonstrations are very helpful to visualize how the material system look like, there is a need for SEM (and/or TEM) of

the pAA-Cys5-drafted silica nanoparticles and membranes used in the demonstration. Also, it is unclear why the membrane-only system shows a cadmium leakage at 60-mL feed water volume - whether this cadmium leakage is due to the membrane integrity or the low loading capacity of pAA-Cys5 on the membrane. If the latter is true, one would be able to address that by increasing the pAA-Cys5 loading on the membrane structure, and systematically study the impacts of pAA-Cys5 loading/density on the Cd ion uptake. While the “membrane + particle” system does show a better performance, we do not understand the main factor that govern such an observation beyond the collective effects of both pAA-Cys5 particles and pAA-Cys5 membranes. Essentially, additional characterization and analyses are essential to elucidate the main factors governing such a performance.

Minor points:

- The term “hyperconfinement” appears to be misused in this work, where basically, the particles are packed in a small volume, and thus polymer chains can be densely packed. Yes, by drafting the copolymers onto silica nanoparticles, the authors can increase the loading (or packing density) of the material per a unit volume, however, stating that the polymers are hyperconfined without a proof of evidence appears to be somewhat overreaching.

- While it is understandable that the regeneration of the material system and release of Cd ion can be challenging and might be within a new scope of study, it is not convincing to state that “the heavy metal and silica particles can be recovered by burning organic materials”. Given the several steps involved in the synthetic process to achieve such a copolymeric system, which may involve several purification steps, burning the organic materials to recover the traces of heavy metal and silica particles might not be helpful in our intensive efforts to minimize global carbon footprint.

- The flow rate of 0.005 liter/min demonstrated in the flow-through membrane system is impractically low.

Reviewer #3 (Remarks to the Author):

The manuscript presents a biomimetic approach to address the global demand for clean water, specifically focusing on the design and application of phytochelatin-inspired copolymers for efficient removal of hazardous heavy metal ions from contaminated waters. The synthesized copolymers, precisely decorated with carboxylate and thiolate moieties, designated as pAA-Cys5, demonstrate exceptional Cd²⁺ ion-capturing capacity in presence of Ca²⁺. By employing detailed characterization techniques and comparing the activity with various synthesized variants, the manuscript elucidates the molecular-level mechanisms underlying polymer-metal ion complex formation. The application of these bio-inspired copolymers in flow-through systems has been demonstrated as highly effective, selectively removing Cd²⁺ ions from heavy metal ion contaminated water and achieving levels suitable for meeting drinking water standards. The study

not only provides valuable insights into the development of water treatment materials but also showcases the potential of biomimetic strategies for environmental remediation.

Technical questions and additional discussion required:

1. Figure 5: It would be valuable for readers if the authors could consider including permeability values for water in this study. This addition would enhance the comparison of the material's performance with other membranes, providing a more comprehensive understanding of its characteristics.
2. It would be beneficial if the authors could include information about the regeneration or cleaning process of the materials, along with insights into the potential loss of adsorption quality over multiple cycles in the flow-through reactor. This addition would contribute to a more comprehensive assessment of the material's cost-effective performance and suitability for potential commercial applications.
3. While the selectivity of pAA-Cys5 for Cd²⁺ ions in the presence of Ca²⁺ ions is highlighted, it would significantly enhance the manuscript's quality if the authors could also demonstrate the selectivity of these materials for other toxic heavy metal ions commonly found in heavy metal ions contaminated environmental water or industrial wastewater. This broader assessment would provide a more comprehensive understanding of the material's applicability across various water treatment conditions.
4. Additionally, in connection with the aforementioned point, incorporating selectivity data for heavy metal ions in the presence of mixed ions, including a mixture of more abundant mono and divalent metal ions along with heavy metal ions to simulate real-world conditions, would not only demonstrate the efficiency of this material but also significantly broaden its potential scope in practical applications.

We thank the reviewers for the careful reviewing and constructive criticisms, which helped us a lot improve our manuscript. As enlisted below, we answered to all the points raised by the reviewers. We carried out a substantial amount of experiments, discussed about these data, and revised the manuscript accordingly.

In what follows, *we repeat the reviewer's comments in red and italics*, we respond to them in green, we cite from the original version of our manuscript in black, and we highlight changes made to the manuscript in blue.

REVIEWER COMMENTS

Reviewer #1 (Remarks to the Author):

In the present work, the authors test polyacrylic acid-based copolymer that they say is inspired by plants. They then examined the ability of the polymer to remove Cd²⁺ from water under continuous flow after coating silica beads and a membrane surface with the polymer. The prepared polymer consists of poly acrylic acids functionalized with cysteine side chains. Notably, they previously reported the same polymer (<https://pubs.rsc.org/en/content/articlelanding/2022/na/d2na00350c>), deposited it on a membrane, showed significant difference in response to Cd over Ca, and showed that the Cd interactions stems from both COOH and SH functionality. Given this, I believe the novelty of the current work stems from demonstrating the materials performance for Cd removal from water, the proposed mechanism for chelation, and the strong affinity for Cd²⁺ over Ca²⁺ unveiled by ITC measurements. In particular, given the high affinity and that mechanistic insight is often missing in the literature, I may be interested in seeing this published in the current journal; however, there are some comments the authors should address below that can help them improve the manuscript. In particular, they should test the material in a competitive environment containing Cd and Ca and many other competing ions commonly found in water sources (and in relevant concentrations) under continuous flow and also provide insight into whether the Cd can be released, and provide all detailed experimental procedures.

1. The authors say that their polymer was inspired by phytochelatin. While it is understood that the two have similar functionality, it is not clear that that functionality is arranged similarly according to figure 1 and so one would expect that the function of the two distinct polymers to be different. Can the authors comment? For instance, is the chelation mechanism for phytochelatin and Cd reported? What are the differences in the binding strength of Cd?

We agree with the reviewer that the binding mechanism of Cd²⁺ to phytochelatin and our polymer

should be discussed more explicitly. To clarify this point, we added the following text block in the revised manuscript together with new references on the structural analysis.

“The chelation mechanism for phytochelatin and Cd^{2+} has been investigated using the combination of different techniques, such as UV/vis, circular dichroism, NMR, mass spectroscopy, potentiometric titration, and isothermal titration calorimetry.^{19, 24-26} Jalilehvand et al. combined Cd K-edge spectra and ^{113}Cd NMR of the concentrated aqueous solution of CdCl_2 (0.5 M) and *N*-acetylcysteine (1.0 M) and proposed the multinuclear complex structures in which the position of Cd is determined by the balance of Cd–S and Cd–O bonds.¹⁴ Wałły et al. investigated the complex stoichiometry of the well-defined oligomers, $(\gamma\text{-Glu-Cys})_n\text{-Gly}$ ($n = 1 - 6$), and determined the chelate stoichiometry as a function of n . They reported the entropic stabilization of the chelate complexes contributes to decrease the total free energy.²⁷”

Regarding the difference in the binding strength of Cd^{2+} ions, we described in the original manuscript by comparing our data and the previously reported data for phytochelatin. As we wrote, it should be noted that the direct comparison of the K_D values determined by different techniques and/or under different measurement conditions is difficult, as we wrote. As we also provided the previously reported values for other materials (Table S1), we believe we have provided sufficient information in order that the readership can compare the binding strength quantitatively. For better clarity, we emphasized that “the other materials” include phytochelatin.

“Although the direct comparison of the K_D values determined by different techniques and/or under different measurement conditions is difficult, the binding affinity of pAA–Cys5 to Cd^{2+} ions is significantly higher than those of previously reported materials. For example, Chekmeneva et al. determined the K_D values of phytochelatin analog $(\alpha\text{Glu-Cys})_4\text{-Gly}$ and glutathione by ITC measurements in 20 mM Tris-HCl buffer containing 0.1 M NaCl (pH 7.4), 3.2×10^{-7} M per molecule and 2.0×10^{-5} M per molecule, respectively.¹⁹ Cheng et al. measured absorption spectra in 0.1 M Tris-HCl (pH 7.4) and obtained a comparable $K_{D(\text{Cd})} = 3.2 \times 10^{-7}$ M for the same phytochelatin analog, $(\alpha\text{Glu-Cys})_4\text{-Gly}$.²⁰ Visvanathan et al. synthesized random peptides mimicking phytochelatin, oligo(*L*-Glu-co-*L*-Cys), and calculated a much weaker affinity to Cd^{2+} ions from the absorption spectra measured in 0.1 M Tris-HCl (pH 7.4), $K_D = 8.6 \times 10^{-4}$ M per molecule.²¹ In this study, the K_D of pAA–Cys5 to Cd^{2+} ions ($K_{D(\text{Cd})} = 2.1 \times 10^{-9}$ M per molecule) was determined in 10 mM Tris-HCl (pH 7.4). **This value is two to five orders of magnitude smaller than the other materials including phytochelatin (Table S1), indicating that bio-inspired synthetic pAA-Cys5 can overtake the function of the naturally occurring peptides $(\text{Glu-Cys})_n$ ”**

2. The authors should report precise procedures for everything. This includes what specifically they did for the preparation of the polymer; including the cysteine grafting, the coating on the silica beads

and membrane preparation. This should include fine details like quantities of starting materials, where starting materials came from, temperatures, solvents, reaction times, etc, so that the work can be reproduced. They should also be sure all details of all experiments used to test the metal ion removal is listed, how materials were regenerated, and tested under continuous flow.

The preparation (synthesis) of polymer (pAA–Cys5) has been reported in our previous account as we wrote in Introduction. For better clarity, we added additional text in the revised manuscript.

“...pAA-based copolymers (pAA–Cys5) possessing –COOH as well as –SH side chains were synthesized (Fig. 1b).¹³⁻¹⁵ In brief: pAA–Cys5 was synthesized by reversible addition–fragmentation chain-transfer (RAFT) radical polymerization of *S*-trityl-cysteine acrylamide (*S*-Tri-Cys-AAm) and acrylic acid (AA) using a radical initiator and a chain transfer agent, followed by deprotection of trityl group with trifluoroacetic acid (TFA). More details can be found in Supplementary Figs. S1-S2.”

The preparation of the polymers used in this study for the first time has been precisely documented in detail in Figs. S6 – S9, S13 – S19, S23 – S27. We checked through the information thoroughly and added some missing information in Supplementary Information such as:

About *the coating on the silica beads*, we added more explicit information in Supplementary Information S20:

“(a) For the column-based microreactor (Figure 4), we functionalized the surface of silica microparticles in three steps. First, a lipid bilayer was deposited on 10 μm-large silica microparticles (300-10 SIL, VDS Optilab) by incubating small unilamellar vesicles of 1,2-dioleoyl-sn-glycero-3-phosphatidylcholine (DOPC) doped with 2 mol% of biotin-DOPE for 60 min (total lipid concentration ≈ 1.3 mM).⁷ Second, the membrane-coated particles were incubated with neutravidin solution (0.6 μM) for 60 min. Finally, the particles were incubated with the solution of pAA–Cys5-biotin (0.5 μM) for 60 min. After each step, the particles were washed three times with PBS buffer.

(b) For the integrative water treatment system (Figure 5), the surface of the silica particle was functionalized with the monolayer of octadecyltrimethoxysilane.⁸ The particles with a diameter of 1.2 μm ((Tokuyama, SS-15)) were incubated with the mixture of DOPC and pAA–Cys5-DOPE (98/2 by mol/mol) dissolved in water/isopropanol for 30 min, then the solvent was step-wisely exchanged to the aqueous buffer following our previous account.⁹”

About the *membrane preparation*, we added more explicit information in Supplementary Information S28:

“Fabrication of integrative water purification system:

The integrative water purification system is based on the combination of two components; hydrophobized silica particles functionalized with the monolayer of octadecyltrimethoxysilane (Fig. S20) and a cellulose membrane coated with chitosan functionalized with *N*-succinimidyl (NHS)-

terminated pAA-Cys5 (chitosan-g-pAA-Cys5, Fig. 5b). A layer of chitosan-g-pAA-Cys5 was deposited on a cellulose membrane ($\Phi = 47$ mm, Advantec) by the filtration of 5 mg/mL dispersion of chitosan-g-pAA-Cys5 in Milli-Q through the membrane using a KG-47 suction system (Advantec). These membrane and beads were packaged into the Amicon cell (Advantec, Fig. S28).”

About the *solvents*, we added more information in Supplementary Information:

Solvent for GPC measurements (in Supplementary Figs. S7, S9): “(10 mM Tris-HCl buffer (pH 7.4) + 100 mM NaCl)”

The *regeneration of the materials* can be achieved via two strategies. In the first strategy, the polymers can be regenerated by cancelling the chelate complex by adding a competitor, such as EDTA (Figure S31). Nevertheless, for the real application, this is to simply concentrate Cd^{2+} ions from one aqueous phase (polluted water model) to another (EDTA solution, Fig. S31). In the second strategy, we proposed that the heavy metal and silica particles can be recovered by burning organic materials.

For the better clarity, we added Supplementary Fig. S31 and explanatory text block as follows.

“The regeneration and recovery of the materials can potentially be achieved by the following two strategies. In the first strategy, the polymers can be regenerated by cancelling the chelate complex by adding a competitor, such as EDTA. As presented in Fig. S31, the continuous treatment with the buffer containing 10 mM EDTA for 200 min resulted in the recovery of function by 83%. Although this suggests the recyclability of the materials, this treatment is to simply concentrate Cd^{2+} ions from one aqueous phase (polluted water model) to another (EDTA buffer). In the second strategy, the heavy metal and silica particles can be recovered by taking the difference in temperature conditions for decomposition into account. For example, the organic compounds can be removed and the metal (Cd) can be recovered in a fluid phase at 600 °C, which is well below the melting temperature of silica (1710 °C). Although the potential recovery of heavy metal can add a unique advantage to our materials over zeolites and other porous inorganic materials, this approach might not be helpful for the minimization of global carbon footprints.”

3. The authors state, “By grafting of bio-inspired copolymers onto silica microparticles and cellulose membranes, 0.1 g (~ 10¹⁷ molecules) copolymers can be confined in ≈ 3 mL volume. Its not clear what the importance of this is, nor what they are referring to when they say 10¹⁷ molecules. Are they trying to state the number of monomers in 0.1 gram of polymer? It’s an awkward way to put things and I am not sure is very relevant.

We partially agree with the reviewer that the way of presentation may sound “*awkward*” to some of the readership. Taking the molecular weight of the polymer ($M_w = 3.7 \times 10^3 \text{ g mol}^{-1}$), 0.1 gram of polymer corresponds to $2.7 \times 10^{-5} \text{ mol}$, i.e. 1.6×10^{17} polymer molecules.

To avoid misunderstanding, we removed “(~ 10¹⁷ molecules)” from Introduction.

We would like to point out that the comparison of the material’s ion loading capacity between different organic, inorganic, and hybrid materials is difficult.

After consulting the literature in the water-treatment field, such as *Chem. Soc. Rev.* **48**, 463-487 (2019), *npj Clean Water* **4**, 36 (2021), we decided to change the documentation of the loading capacity by “milligram of captured ions per 1 gram of the material”. The changes were reflected in the revised manuscript.

4. To improve the readers ability to follow what they are doing can they also provide the concentration of the metal ions in ppm and ppb rather than molarity or at least both?

We agree with the reviewer and presented the concentrations of metal ions both in molarity and ppm (Figs. 3b, 3c, 3d, 4c, and 5d). In addition, the cumulative amount of captured Cd²⁺ is also indicated in mg (Fig. 4d).

To clarify this point, we added the following text to caption to Fig. 3:

“Note that [Cd²⁺] for the WHO’s drinking water standard (0.03 μM) corresponds to 0.003 ppm.”

5. They talk about the polymers performance first, but it’s a little misleading as the polymers must be immobilized onto other particles for water purification applications. So, to look at the performance of the polymer by mass is misleading in my opinion. So, maybe if they state specifically why they did this study upfront with only the polymer, albeit not possible to use in this format for water purification studies, it would be helpful. The reason I say this is because at the moment, the section seems rather irrelevant.

As we answered to Point 1, the assessment and comparison of the material's performance is only possible by characterizing the materials in solution. For example, the binding affinity of pAA–Cys5 to Cd^{2+} and those of phytochelatin and other materials can only be compared by measuring ITC in solution, because there have been no data on immobilized phytochelatin.

To clarify this point, we added the following text:

“To verify the applicability of the design strategy in the synthesis of plant phytochelatin-inspired materials, we determined the binding affinity of pAA–Cys5 to Ca^{2+} and Cd^{2+} ions by measuring ITC in solution and compared the affinity with those of phytochelatin and other materials.”

6. Its recommended that the authors include more controls in their study. They should consider looking at the performance of the non-functionalized PAA, PAA-silica, and PAA-cellulose membrane, and just the cellulose membrane, along with the ones they have included (PAA-Cy5, the PAA-Cy-5-silica, and PAA-Cy5-cellulose membrane).

Following the reviewer's comment, we present the ITC data of pAA with Ca^{2+} and Cd^{2+} in Supplementary Fig. S3, which clearly indicate that pAA does not interact with Cd^{2+} ions.

For a better clarity, we modified the manuscript as:

“In contrast, the binding affinities of pAA (devoid of cysteine side chains) to Ca^{2+} and Cd^{2+} ions are too low to calculate the K_D values from the ITC data (Fig. S3)”.

Another control experiment on unmodified cellulose membranes was performed using the integrative water treatment systems, showing a significant leakage from the very first fraction (Fig. 5d).

This is now explained in the revised manuscript:

“In the presence of unmodified cellulose membrane and silica microparticles without polymers, the system could not remove Cd²⁺, showing significant leakage from the very first fraction (Fig. 5d).”

7. The authors state: “As shown in Fig. 3a, the pAA–Cys5 solution and the buffer containing Cd²⁺ were allowed to react overnight, and the [Cd²⁺] in the ultrafiltrated flow-through was determined using inductively coupled plasma optical emission spectrometry (ICP-OES)” Its really unclear to me what they mean by ultrafiltrated flow through....are they intending to say that they are using ultrafiltration to remove the cadmium from the solution that contains suspended polymers? If they give a more precise description here it would be helpful as what they are talking about as this is not commonly encountered in water purification studies. I had to read this section multiple times and stare at the image a bit to really follow what they are doing.

We thank the reviewer for raising this important point. To clarify the point raised by the reviewer “*Its really unclear to me what they mean by ultrafiltrated flow through*”, we added the explanatory text in the revised manuscript:

“... As shown in Fig. 3a, the pAA–Cys5 solution and the buffer containing Cd²⁺ were allowed to react overnight, followed by ultracentrifugation. Because the pAA–Cys5 polymer with $M_w \approx 1.7 \times 10^4$ Da cannot pass the filter with the cut-off level of 3×10^3 Da, only the ions that did not bind to pAA–Cys5 can pass through the filter. The [Cd²⁺] in ...”

8. Along the same lines above, Figure 3 a is not clear. They need to provide some kind of labels to guide the reader in what they are looking at. After looking at it a bit, I finally think I understand, but it should not really be this difficult.

Following the reviewer’s comment, we modified Fig. 3a and added (i) the average M_w of the polymers are ≥ 17 kDa, and (ii) the cut-off M_w of the filter is 3 kDa.

Accordingly, the caption to Fig. 3a was modified as:

“Only the ions that did not bind to pAA-Cys5 can pass through the filter because the ions bound to pAA-Cys5 polymer ($M_w \approx 1.7 \times 10^4$ Da) cannot pass the filter.”

9. What is the molecular weight of the polymer they are producing? Why is it that they cannot make polymeric beads, as found in resins, rather than putting it onto a silica support or onto a membrane?

The M_w and M_w/M_n values of the pAA-Cys5 biotin were $M_w = 1.7 \times 10^4$ Da and $M_w/M_n = 2.2$, as presented in Fig. S19. We also presented the corresponding values for the pAA-Cys5-DOPE, $M_w = 7.1 \times 10^3$ Da and $M_w/M_n = 1.9$ (Fig. S25).

As we reported previously (Tutus, et al., *Adv. Funct. Mater.* **22**, 4873-4878 (2012)), the use of silica particles with a higher density is advantageous for the spatial confinement of the polymers in a smaller volume. In contrast, if we cross-link and fabricate pAA-Cys5 beads, the density of hydrated polymer beads becomes very close to that of aqueous medium. Therefore, the packaging of the polymer beads by sedimentation is practically not possible.

To clarify this point, we modified the revised manuscript:

“... Note that the grafting of pAA-Cys5 brushes on silica particles have two advantages. First, as demonstrated previously,³⁸⁻⁴¹ the average distance between polymer chains $\langle d \rangle$ can be controlled by

the molar fraction χ at a nm-accuracy, $\langle d \rangle = \sqrt{A_{\text{lipid}}/\chi}$, where A_{lipid} is the cross-sectional area of one lipid molecules, ($A_{\text{lipid}} \approx 0.6$ nm²).⁴² For example, the inter-polymer distance corresponding to $\chi = 0.02$ can be estimated as $\langle d \rangle \approx 5.5$ nm. Second, the grafting of brushes on silica particles makes the separation of the polymer-Cd²⁺ complex much simpler. The use of pAA-Cys5 gels / particles is not practically possible, because the density of the hydrated polymers is very close to the density of the medium. The homogeneous coating of ...”

10. The authors change their experiment when they look at pAA-Cys5 in the presence of cadmium only

and then with cadmium and calcium, they change their experiment. It would be good if they can do it the same way (0.01mg/mL pAA-Cys5) and then 10mM Cd²⁺ and 500mM Ca²⁺. Then figure 3b and 3d could be combined. Also, where are the error bars on 3d? They should be sure to provide error bars (and esds in the text) when possible, for all experiments.

We thank the reviewer for the careful reviewing. After reading the comment, we found that the polymer concentration we wrote in the caption to Figure 3b (1.0 mg mL⁻¹) was wrong and not consistent with what we wrote in the main text (0.1 mg mL⁻¹). We corrected the caption to Fig. 3b. Therefore, the data presented in Figs. 3b – 3d were all acquired at the same polymer concentration (0.1 mg mL⁻¹), indicating that all the experiments were performed under the same conditions.

In the revised manuscript, we correct the caption to Figure 3b as:

“(b) [Cd²⁺] in the flow-through in the absence (light gray) and presence (red) of pAA–Cys5 (0.1 mg mL⁻¹).”

Following the reviewer’s comment, we added the error bars to Fig. 3d, taking the calibration accuracy of Spectroquant® colorimetric assay. The caption to Fig. 3d reads “Error bars indicate the calibration accuracy of Spectroquant® colorimetric assay.”

The data presented in Figure 2 and Figure 3b were already shown with the error bars. We also added the error bars to Figs. 4c and 5d. The errors in X axis originate from the fraction volume, whereas those in Y axis from the calibration accuracy of Spectroquant® colorimetric assay.

11. Along the same lines of comment 10, they are adsorbing a lot of Ca^{2+} because the concentration of Ca^{2+} is much higher than Cd. Thus, I think they have as much Ca or even more than Cd in the polymer, despite the higher affinity towards Cd^{2+} . For example, they remove almost 50% of the Ca^{2+} concentration at a polymer loading of 0.1 mg/mL. If they see no change in the IR spectrum associated with the polymer in the presence of Ca^{2+} , then what do they suppose is the nature of the interaction between Ca and their polymer? Given this, I would like to see how the Ca and other ions impact the materials performance in the final flow through experiment, like Na, K, Ca, among others.

We thank the reviewer for raising this point. As the reviewer commented, we also think it is reasonable that Ca^{2+} interact with pAA-Cys5 at $[\text{Ca}^{2+}] = 0.5 \text{ mM}$ ($5 \times 10^{-4} \text{ M}$), because the K_D value of pAA-Cys5 and Ca^{2+} is $\sim 10^{-5} \text{ M}$. However, the interaction could not be detected by FTIR at high concentrations ($\sim 10^{-3} \text{ M}$, Figs. 2c and 2d) because only a small portion of $-\text{COO}^-$ groups contributes to the spectral signals, because the fraction of Cys side chains is only 5 mol%. In fact, the change in the spectral intensity could be detected only in differential spectra even in the presence of $[\text{Cd}^{2+}] \sim 10^{-3} \text{ M}$, which is about six magnitudes larger than the K_D value ($\sim 10^{-9} \text{ M}$).

To clarify this point, we present the data on Ca^{2+} in Fig. S11 (note that the data presented in Fig. 3d was replaced by the data measured in the presence of Na^+ , K^+ , Mg^{2+} , and Ca^{2+}), added the ITC data for Na^+ , K^+ , and Mg^{2+} (Fig. S12), and added the explanatory text in the revised manuscript:

“As presented in Fig. S11, we also detected the decrease in $[\text{Ca}^{2+}]$ in flow-through, suggesting that Ca^{2+} interact with pAA-Cys5. This is reasonable at $[\text{Ca}^{2+}] = 0.5 \text{ mM}$ ($5 \times 10^{-4} \text{ M}$), which is more than one order of magnitude higher than the K_D value of pAA-Cys5 and Ca^{2+} ($\sim 10^{-5} \text{ M}$). As presented in Fig. S12, the interactions of pAA-Cys5 with Na^+ , K^+ , and Mg^{2+} are much weaker. Notably, the interaction of pAA-Cys5 and Ca^{2+} could not be detected by FTIR spectra, which were measured even at higher concentrations ($\sim 10^{-3} \text{ M}$, Figs. 2c and 2d). This finding can be explained by the molar fraction of Cys side chains (5 mol%). Namely, only a small portion of $-\text{COO}^-$ groups contributes to the spectral signals. In fact, the change in the spectral intensity could be detected only in differential spectra even at $[\text{Cd}^{2+}] \sim 10^{-3} \text{ M}$, which is about six magnitudes larger than the K_D value ($\sim 10^{-9} \text{ M}$). Therefore, it seems reasonable that we could not detect the interaction of pAA-Cys5 with Ca^{2+} and other cations spectroscopically.”

To address the next point raised by the reviewer, “*I would like to see how the Ca and other ions impact the materials performance in the final flow through experiment, like Na, K, Ca, among others*”, we performed the experiments in the presence of $[Cd^{2+}] = 0.01 \text{ mM}$ ($10 \text{ }\mu\text{M}$), $[Na^+] = 1 \text{ mM}$, $[K^+] = 0.2 \text{ mM}$, $[Mg^{2+}] = 0.5 \text{ mM}$, and $[Ca^{2+}] = 0.5 \text{ mM}$, which simulates the concentration of the abundant ions in ground water. As presented in the new Fig. 3d, 0.1 mg/mL of pAA–Cys5 polymers could remove Cd^{2+} to the level below WHO’s drinking water standard, $< 0.03 \text{ }\mu\text{M}$ (0.003 ppm).

In the revised manuscript, we presented the new Fig. 3d and modified the text:

“To test if pAA–Cys5 can selectively capture Cd^{2+} , we incubated pAA–Cys5 solutions (10^{-4} , 10^{-3} , 10^{-2} , and $10^{-1} \text{ mg mL}^{-1}$) with a buffer containing $[Cd^{2+}] = 0.01 \text{ mM}$ ($10 \text{ }\mu\text{M}$), $[Na^+] = 1 \text{ mM}$, $[K^+] = 0.2 \text{ mM}$, $[Mg^{2+}] = 0.5 \text{ mM}$, and $[Ca^{2+}] = 0.5 \text{ mM}$, which mimic the concentration levels in wastewater, respectively.³⁴⁻³⁶ Fig. 3d shows $[Cd^{2+}]$ in flow-through, indicating that pAA–Cys5 can capture Cd^{2+} even in the presence of a large excess of abundant mono- and divalent metal ions in ground water.

When using 0.1 mg mL^{-1} of pAA–Cys5, $[Cd^{2+}]$ in flow-through was lower than the acceptable concentration declared by WHO for drinking water ($< 0.03 \text{ }\mu\text{M}$). These results demonstrated that the polymer could be used to selectively remove Cd^{2+} but not other ions in groundwater, such as Na^+ , K^+ , Mg^{2+} , and Ca^{2+} . Such a high selectivity to toxic Cd^{2+} ions make them distinct from zeolites and ion exchange resins, because the ions possessing similar sizes and charges are equally captured.”

12. In the experiments shown in Table 3C. Why is it that when they decrease the pAA-Cys5 loading to 0.001 they see a decrease in capacity? This does not make any sense to me. Should it not be as high as that observed at 0.01 or higher? If there are esds associated with these measurements.

We thank the reviewer for raising this point. We agree that this part should be explained more clearly.

First of all, we want to emphasize that the numbers presented in Table 3c are the amount of Cd^{2+} ions to 1 g of polymer, which is different from the “maximum loading capacity”.

To clarify this point, we replaced the terminology in the legend “loading capacity” to “ Cd^{2+} bound to 1 g pAA–Cys5” and corrected the caption to “(c) Table showing the initial concentrations of pAA–Cys5 and Cd^{2+} ($[pAA-Cys5]$ and $[Cd]_{\text{initial}}$, respectively) and Cd^{2+} in the flow-through ($[Cd]_{\text{flowthrough}}$). These

values were used to calculate the amount of Cd²⁺ bound to 1 g of pAA–Cys5,”.

[pAA-Cys5] [mg/mL]	[Cd] _{initial} [μM]	[Cd] _{flow-through} [μM / ppm]	Cd / 1 g pAA-Cys5 [mg/g]
0.1	10	<0.02 / <0.002 ^a	–
0.1	100	0.6 / 0.06 ^a	105.3
0.01	100	42.8 / 4.79 ^b	582.4
0.001	100	92.0 / 10.3 ^b	336.0
0 (control)	100	95.0 / 10.6 ^b	–

^aSpectroquant cadmium test, ^bICP-OES.

Second, the amount of bound Cd²⁺ ions are determined by two counteracting effects depending on the polymer concentration. At high polymer concentrations, the amount of bound ions becomes less with increasing polymer concentration, exactly as the reviewer commented. On the other hand, at low polymer concentrations, the equilibrium between “bound” and “unbound” states shifts towards the unbound state. Namely, this value should take a maximum based on the balance of these two effects. This point is explicitly explained in the revised manuscript:

“... (Fig. 3c). It should be noted here that at high polymer concentrations, the amount of bound ions per polymer becomes less with increasing polymer concentrations, because the binding sites in polymers are not saturated by Cd²⁺. On the other hand, at low polymer concentrations, the equilibrium between bound and unbound states shifts towards the unbound state. Therefore, the loading capacity of pAA-Cys5 shows maximum at [pAA-Cys5] = 0.01 mg/mL based on the balance of these two effects. The maximum loading capacity of...”

13. The authors state, “The estimated maximum loading capacity per 1 g of polymer was 11 mmol g⁻¹.” It should instead be “1 gram of polymer is 11 mmol”. Also, considering that the polymer cannot be employed here as the active material alone and they need a support, they should also report the total mass of the coated beads that would be required to remove 11 mmol of Cadmium. Otherwise, it is deceiving.

The first comment seems to be a misunderstanding. We explicitly mean “11 mmol of Cd²⁺ ions bind to 1 g of polymer” to compare the binding affinity with the other materials. We agree that the binding amount for 1 mol of polymer should be presented.

Following the reviewer’s suggestion, we provided the *total mass of the coated beads* in the revised manuscript:

“Here, 8.2 × 10⁻⁴ g, corresponding to 4.9 × 10⁻⁸ mol, of polymers were grafted to 0.85 g of silica microparticles. This yields the estimated maximum loading capacity of Cd²⁺ per 1 g of polymer to be 11 mmol g⁻¹. Taking the number-average molecular weight (*M_n*) of pAA–Cys5 (8.0 × 10³) and pAA–Cys5-biotin (7.4 × 10³), the difference in the maximum loading capacity of polymers with and without confinements are estimated to be 41.6 mol mol⁻¹ and 79.8 mol mol⁻¹, respectively.”

14. I think they should get rid of these small cartoons they are using in figure 3d unless they modify it to make it clearer to the reader what they are looking at.

We agree with the reviewer that this schematic illustration was too small. As we still believe this illustration helps the understanding of the readership who are not familiar with the water treatment systems, we modified the layout of Figure 3 for a better visual clarity. In case this is still hard to understand, we are willing to modify the illustration and layout.

15. The authors state “The obtained capacity is two times higher than that recorded in the solution system (5.2 mmol g⁻¹), which can be attributed to the fact that the polymers were confined within a small volume.” Its not clear why the capacity of the material on a per gram basis would increase with respect to the volume. Its very awkwardly written. While the sentence after it clarifies, they should rewrite this to be more concise.

We thank the reviewer for pointing out this aspect. To clarify the point, we added the following text block with additional references and discussed about the effect of hyper-confinement (neighbor effect or multivalent binding).

“Currently, we interpret the higher loading capacity realized by the particle-grafted pAA-Cys5-biotin in terms of the neighbor effect or the multivalent binding.⁴³⁻⁴⁵ The former is a well-established concept in understanding the effect of adjacent ligands on the binding to a linear lattice. The latter is the principle of a targeted strengthening of interactions between the binding partners forming cooperative, multiple interactions that are based on individually weak, noncovalent bonds. In both cases, the presence of binding partners in the vicinity strengthens the overall binding. As suggested by the NMR analysis, the determination of valency of interaction, i.e. the number of contributing moieties in the chelate formation, is difficult in our experimental systems. Nevertheless, it is plausible that Cd²⁺ that

escapes from one complex can readily form another complex by interacting with a neighboring –CH₂S⁻ and –COO⁻ pairs. ...”

16. Can the authors comment on why it takes about 24 h to treat 36 mL of water? Why are they using such low flow rates, what are the pressures? Could this not be overcome by using larger silica beads rather than the 10 micron ones. I think you can buy commercial silica beads with much larger diameters? Or is there some other reason that the flow was so low? More explanations could help readability.

We thank the reviewer for raising this key point.

Previously, we fabricated a flow-through biochemical microreactor by packaging sarcoplasmic reticulum membrane deposited on silica microparticle with a diameter of 10 μm in a chromatography column with an inner diameter of 10 mm [Tutus, et al., *Adv. Funct. Mater.* **22**, 4873–4878 (2012)]. For the first prototype (Figure 4), we selected the same particle size and the column size, which resulted in the flow rate of 1.5 mL/h, which was about one half of the previous study.

In general, the technical limit of the flow rate is defined by the back pressure P , which follows:

$$P \propto \frac{L \times F}{D^2 \times ID^2}.$$

L is the length of the path (packed beads), F the flux, D the particle diameter and ID is the inner diameter of the column. The use of larger particles decreases the pressure if L is kept constant, e.g. the use of 20 μm-large particles reduces the P by a factor of 4. However, in order to keep the same surface area, one needs 2 times larger bed volume and hence the length L .

Therefore, we concluded that the prototype was useful to demonstrate the proof of principle but not suited for achieving a realistic flow rate comparable to a fixed-bed column system [*ChemBioEng Rev.* **5**, 173-179 (2018)].

In the revised manuscript, we added more precise explanation:

“... in practice. This extremely low flow rate is caused by a high back pressure P , which scales with:

$$P \propto \frac{L \times F}{D^2 \times ID^2}.$$

L is the length of the path (packed beads), F the flux, D the particle diameter and ID is the inner diameter of the column. The use of larger particles decreases the pressure if L is kept constant, e.g. the use of 20 μm-large particles reduces the P by a factor of 4. However, in order to keep the same surface area, one needs 2 times larger bed volume and hence the length L . Therefore, we concluded that the prototype was useful to demonstrate the proof of principle but not suited for achieving a realistic flow rate comparable to a fixed-bed column system.⁴⁶”

17. The section on increasing the efficiency of their flow through setup is confusing. They keep saying they are increasing the area to volume ratio but never discuss what they are increasing the area to volume ratio of. I am assuming they are simply trying to increase the quantity of polymer by using smaller silica particles which offer a larger surface area to volume ratios. They need to revise this section as its very hard for readers to follow.

The consideration of area -to-volume ratio is reciprocal, because the decrease in the particle diameter by a factor of 8 results in the increase in the area-to-volume ratio by a factor of $512/64 = 8$.

For the clarity, we added a text block explaining this point.

“... small particles. For example, the decrease in the particle diameter by a factor of 8 results in the increase in the area-to-volume ratio by a factor of $512/64 = 8$. However, as described above, the use of small particles results in a significant increase in the back pressure P , because $P \propto D^{-2}$. Therefore, to realize both a higher area-to-volume ratio and a lower back pressure, we designed a new reactor.”

18. Why did they change the coating process when they reduced the size of the silica beads?

We thank the reviewer for raising this point.

We changed the coating protocol for smaller particles (diameter 1.2 μm) to minimize the risk of particle aggregation. When we reduced the size of silica microparticles, we found a marked aggregation of particles after each centrifugation. Once the beads form aggregates, they could hardly be re-suspended as single particles. To reduce the centrifugation/washing cycles, we deposited a lipid monolayer incorporating pAA–Cys5–DOPE (2 mol%) on the surface of silanized silica microparticles in one step, following our previous account [*Nanoscale Adv.* **4**, 5027-5036 (2022)]. The covalent coupling of a pAA–Cys5 chain directly to the lipid head group also ensure the stability of the polymer chains, which was another positive effect.

To clarify this point, we added an explanatory text block and a new Fig. S20:

“...particle aggregation. Once the beads form aggregates, they could hardly be re-suspended as single particles. The surface functionalization with a lipid monolayer incorporating pAA–Cys5–DOPE (2 mol%) following our previous account⁴⁷ prevented the undesired aggregation of particles caused by the multiple centrifugation cycles (Fig. S20). The covalent coupling of a pAA–Cys5 chain directly to the lipid head group also ensure the stability of the polymer chains, which was another positive effect. The molecular sieve...”

19. The authors state, “The molecular sieve (pore diameter: 0.1 μm) used in the first prototype (Fig. 4) was replaced by a cellulose membrane coated with chitosan functionalized with N-succinimidyl (NHS)-terminated pAA– Cys5 (chitosan–g–pAA–Cys5, Fig. 5b) to achieve high flux under conditions

of low back pressure.

First, I am assuming the membrane they use somehow provides a much better flow rate over the molecular sieve. If that is the case, they should clearly state this.

We agree with the reviewer that this is an important point. In fact, in the original manuscript, we already described:

“Notably, the new system could be operated at a flow rate of 5 mL min⁻¹, **which is 250 times higher than the flow rate used for the previously described prototype** (Fig. 4).”

To clarify this point, we modified the text as:

“...(chitosan-g-pAA-Cys5, Fig. 5b) to **reduce the back pressure and hence to increase the flow rate.**”

Second, why do the authors prepare a membrane coated with pAA-Cys5 as well as the coated beads in the same separation unit? I am assuming the membrane is just there to inhibit the loss of their beads. They should also do a control here to show that the Cd goes through the non-functionalized membrane. I did not see this, unless I missed it, and do the experiment with only the functionalized membrane as well.

As the reviewer pointed out above, the replacement of a molecular sieve by a cellulose membrane led to a significant increase in the flow rate (by a factor of 250). To minimize the risk of Cd²⁺ leakage under such a high flow rate, we utilized a cellulose/chitosan-pAA-Cys5 membrane.

(1) To answer the reviewer’s comment, we performed a control experiment “*to show that the Cd goes through the non-functionalized membrane*”.

(2) Following the reviewer’s comment, “*the experiment with only the functionalized membrane*” was carried out.

Both data sets are presented in Figure 5d in the revised manuscript. For the clarity, we changed the Figure legend and caption.

“The unmodified cellulose membrane and silica microparticles without polymers showed a significant leakage from the very first fraction (light gray). The cellulose/chitosan-g-pAA-Cys5 membrane

showed a leakage of Cd^{2+} already at $V = 60$ mL (gray). In contrast, the combination of the membrane and the pAA–Cys5-functionalized microparticles (black) could remove $10 \mu\text{M}$ Cd^{2+} (typical industrial wastewater level) from 0.3 L in 1 h down to the level approved for drinking water.”

20. Notably, 5 grams of material (beads and polymer, I presume) to purify 0.3 L of water is a lot, which may be further optimized later. So, they should also describe the overall mass of polymer used in this flow through setup. What is the polymer/bead mass ratio in this case?

Following the reviewer’s suggestion, we provided the *overall mass of polymer and polymer/bead mass ratio* in the revised manuscript:

“Here, 0.063 g, corresponding to 8.9×10^{-6} mol, of pAA–Cys5–DOPE deposited on 5 g of silica microparticles yielding the polymer/bead mass ratio of 0.013. We monitored...”

21. The state that that the polymer can be burned off to recover the cadmium. To state this, they should give protocols, unless I missed it for how the cadmium is retrieved and then the methods to clean the silica surface and recoat the beads and report the performance after bead reuse.

Our suggestion towards the potential recovery of Cd is based on the differential melting/boiling points of organic compounds, metals and silica. The melting point and the boiling point of silica are $1710 \text{ }^\circ\text{C}$ (1984 K) and $2230 \text{ }^\circ\text{C}$ (2504 K), respectively. The corresponding values for Cd are $321 \text{ }^\circ\text{C}$ (595 K) and $765 \text{ }^\circ\text{C}$ (1039 K), respectively. As organic materials are supposed to be burned out at $600 \text{ }^\circ\text{C}$ (*RSC Adv.* **5**, 42572-42579 (2015)), we assume that the heating of the systems to $600 \text{ }^\circ\text{C}$ results in solid silica and melted, liquidous Cd. To avoid the confusion, we added explanatory text about the outline of possible *protocols* and point out the advantage and disadvantage of this strategy.

“The regeneration and recovery of the materials can potentially be achieved by the following two strategies. ... In the second strategy, the heavy metal and silica particles can be recovered by taking the difference in temperature conditions for decomposition into account. For example, the organic compounds can be removed and the metal (Cd) can be recovered in a fluid phase at $600 \text{ }^\circ\text{C}$, which is well below the melting temperature of silica ($1710 \text{ }^\circ\text{C}$). Although the potential recovery of heavy metal can add a unique advantage to our materials over zeolites and other porous inorganic materials, this approach might not be helpful for the minimization of global carbon footprints.”

22. I would also be interested to know if they can regenerate the polymer using other chemical means, or is the Cd so strongly bound that one must pyrolyze the material?

We thank the reviewer for raising this point. We have confirmed that the polymers can be regenerated

by eluting EDTA solution, and the data are presented in Fig. S31. Nevertheless, it should be noted that this regeneration procedure enables to reuse polymers, but Cd^{2+} ions are just transferred to another aqueous solution (flow through). In the revised manuscript, we explained these additional data and discussed about the advantage and disadvantage of this approach.

“The regeneration and recovery of the materials can potentially be achieved by the following two strategies. In the first strategy, the polymers can be regenerated by cancelling the chelate complex by adding a competitor, such as EDTA. As presented in Fig. S31, the continuous treatment with the buffer containing 10 mM EDTA for 200 min resulted in the recovery of function by 83%. Although this suggests the recyclability of the materials, this treatment is to simply concentrate Cd^{2+} ions from one aqueous phase (polluted water model) to another (EDTA buffer).”

23. The authors should work on the overall paper flow. It was difficult for me to read and hence I had to go over it several times.

We thank the reviewer’s comment. We went over the manuscript, consulted a native but non-expert reader, improved the connection between the sections, and add more explicit discussions for a better clarity.

Reviewer #2 (Remarks to the Author):

In this work, phytochelatin-inspired copolymers containing carboxylate and thiolate moieties were synthesized, and grafted onto silica nanoparticles and cellulose membranes for the removal of Cd ion from contaminated water in a flow-through system. The copolymers consisting of poly(acrylic acid) partially functionalized with cysteine side chains were synthesized via reversible addition-fragmentation chain-transfer (RAFT) radical copolymerization of acrylic acid (AA) and AA coupled to cysteine, and was referred to as pAA-Cys5. It was shown that the cadmium removal effectiveness (representing by the dissociation constant for Cd ion) of pAA-Cys5 is four-to-five orders of magnitude higher than that for peptides mimicking the sequence of endogeneous phytochelatin. Isothermal titration calorimetry, FTIR, and NMR spectroscopy were used to elucidate the structures of the polymer-ion complexes.

In general, the work represents important efforts in selective removal of cadmium from contaminated water to provide drinking-water with acceptable traces of cadmium. Control experiments and characterization were carefully conducted to elucidate the main complexation mechanism between Cd ions and pAA-Cys5. For example, two other copolymers with the main chains containing monomers devoid of carboxyl groups (i.e., pHPMA-Cys5 and pPEGMA-Cys5) were tested to access the role of carboxyl groups on PAA in complexation with Cd ion. It was hypothesized and empirically confirmed that cadmium ion forms a complex with one Cys chain and carboxyl group(s) of the PAA main chain. While interesting insight is provided, from the reviewer's perspective, fundamental insights and engineering breakthrough gathered from this work are not at the level of novelty suitable for publishing on Nat. Commun.

We disagree with the reviewer's comment, "cadmium ion forms a complex with one Cys chain and carboxyl group(s) of the PAA main chain" was "hypothesized and empirically confirmed".

As we presented in Fig. 2, the FTIR data provided direct evidence that the presence of Cd²⁺ modulated antisymmetric and symmetric stretching of –COO[–] (Figs. 2c and 2d). We also measured the NMR spectra of model compound NAC, because no clear feature can be extracted from polymer (Fig. S2). As presented in Fig. 2e, the NMR spectra of NAC demonstrated that Cd²⁺ forms a complex with –COO[–] and –CH₂S[–] groups. The possible structures of the Cd²⁺–pAA–Cys5 complex deduced from the NMR and FTIR spectra were presented in Fig. 2f.

Moreover, the data from "two other copolymers with the main chains containing monomers devoid of carboxyl groups (i.e., pHPMA-Cys5 and pPEGMA-Cys5)" as well as "pAA devoid of cysteine side chains" provided supporting evidence that "the coexistence of both moieties is essential".

Fundamentally, while interesting, this work does not address the main challenges or scientific gaps encountered in this field of separation, which involve 1) the separation between similar ions, or 2) the

separation of a target ion, such as Cd ion, from a complex multi-ion mixture. While the authors claim that the proposed material system offers “specific recognition functions of biological systems” (page 16), the ion specificity/recognition capability of pAA-Cys5 toward Cd ion has not been well-understood nor demonstrated in-depth. Merely testing the separating performance of pAA-Cys5 in a mixture of two divalent cations with different ion properties (e.g., Cd and Ca ions) does not sufficiently demonstrate the ion recognition and discrimination capability of the material.

We thank the reviewer for raising this important point. In fact, a similar comment was also raised by the reviewer 1.

To directly answer the comment “separation from a complex multi-ion mixture”, we performed additional experiments in the coexistence of $[K] = 10 \mu\text{M}$, $[Na] = 1000 \mu\text{M}$, $[K] = 200 \mu\text{M}$, $[Mg] = 500 \mu\text{M}$, and $[Ca] = 500 \mu\text{M}$, which is a realistic model of the ions in the ground water.

In the revised manuscript, we replaced the scheme and the data in Fig. 3d by the data for the complex multi-ion mixture.

We also added the following text in the revised manuscript:

“To test if pAA-Cys5 can selectively capture Cd^{2+} , we incubated pAA-Cys5 solutions (10^{-4} , 10^{-3} , 10^{-2} , and $10^{-1} \text{ mg mL}^{-1}$) with a buffer containing $[Cd^{2+}] = 0.01 \text{ mM}$ ($10 \mu\text{M}$), $[Na^+] = 1 \text{ mM}$, $[K^+] = 0.2 \text{ mM}$, $[Mg^{2+}] = 0.5 \text{ mM}$, and $[Ca^{2+}] = 0.5 \text{ mM}$, which mimic the concentration levels in wastewater, respectively³⁴ Fig. 3d shows $[Cd^{2+}]$ in flow-through, indicating that pAA-Cys5 can capture Cd^{2+} even in the presence of a large excess of abundant mono- and divalent metal ions in ground water. When using 0.1 mg mL^{-1} of pAA-Cys5, $[Cd^{2+}]$ in flow-through was lower than the acceptable concentration declared by WHO for drinking water ($< 0.03 \mu\text{M}$). These results demonstrated that the polymer could be used to selectively remove Cd^{2+} but not other ions in groundwater, such as Na^+ , K^+ , Mg^{2+} , and Ca^{2+} . Such a high selectivity to toxic Cd^{2+} ions make them distinct from zeolites and ion exchange resins, because the ions possessing similar sizes and charges are equally captured. As presented in Fig. S11, we also detected the decrease in $[Ca^{2+}]$ in flow-through, suggesting that Ca^{2+} interact with pAA-Cys5. This is reasonable at $[Ca^{2+}] = 0.5 \text{ mM}$ ($5 \times 10^{-4} \text{ M}$), which is more than one order of magnitude higher

than the K_D value of pAA–Cys5 and Ca^{2+} ($\sim 10^{-5}$ M). As presented in Fig. S12, the interactions of pAA–Cys5 with Na^+ , K^+ , and Mg^{2+} are much weaker. Notably, the interaction of pAA–Cys5 and Ca^{2+} could not be detected by FTIR spectra, which were measured even at higher concentrations ($\sim 10^{-3}$ M, Figs. 2c and 2d). This finding can be explained by the molar fraction of Cys side chains (5 mol%). Namely, only a small portion of $-\text{COO}^-$ groups contribute to the spectral signals. In fact, the change in the spectral intensity could be detected only in differential spectra even at $[\text{Cd}^{2+}] \sim 10^{-3}$ M, which is about six magnitudes larger than the K_D value ($\sim 10^{-9}$ M). Therefore, it seems reasonable that we could not detect the interaction of pAA–Cys5 with Ca^{2+} and other cations spectroscopically.”

Furthermore, we examined the capacity of our integrated system to separate the complex multi-ion mixture, too. The $[\text{Cd}^{2+}]$ level remained below the detection limit of the colorimetric assay up to 135 mL at a flow rate of 5 mL min^{-1} (open symbols, Fig. S30).

In the revised manuscript, we clarified this point:

“...within 1 h. The Cd^{2+} removal capacity of our system was further examined by treating the simulated wastewater containing $[\text{Cd}^{2+}] = 0.01 \text{ mM}$ (10 μM), $[\text{Na}^+] = 1 \text{ mM}$, $[\text{K}^+] = 0.2 \text{ mM}$, $[\text{Mg}^{2+}] = 0.5 \text{ mM}$, and $[\text{Ca}^{2+}] = 0.5 \text{ mM}$. The $[\text{Cd}^{2+}]$ level remained below the detection limit of the colorimetric assay up to 135 mL at a flow rate of 5 mL min^{-1} (Fig. S30).”

Even among the separation of these two divalent cations, it is unclear why pAA-Cys5 has a stronger affinity toward Cd ion than Ca ion. While the authors successfully elucidated the mechanism at which

Cd ion form a complex with the pAA-Cys5 system, the main mechanism at which the two divalent cations are discriminated by pAA-Cys5 is not discussed. In-depth characterization and discussion are essential to provide more insight governing the behaviors of pAA-Cys5.

It is unclear what the reviewer meant by the comment “*the main mechanism at which the two divalent cations are discriminated by pAA-Cys5 is not discussed*”. First, the four orders of magnitude difference in the dissociation constants K_D was quantitatively determined using ITC (thermodynamics, Figs. 2a and S12). The structure of the Cd^{2+} -pAA-Cys5 complex was elucidated by the combination of FTIR and NMR (Figs. 2b – 2f). In contrast, no structural feature of the Ca^{2+} -pAA-Cys5 complex could be detected spectroscopically. Moreover, we demonstrated that the coexistence of carboxylate in pAA main chain and thiolate in cysteine side chain is necessary by using the polymer devoid of each moiety (Figs. S3 and S10).

As described in our answer to the previous comment, we carried out additional experiments and discussed about the differential interactions, such as:

“Fig. 3d shows $[\text{Cd}^{2+}]$ in flow-through, indicating that pAA-Cys5 can capture Cd^{2+} even in the presence of a large excess of abundant mono- and divalent metal ions in ground water. When using 0.1 mg mL^{-1} of pAA-Cys5, $[\text{Cd}^{2+}]$ in flow-through was lower than the acceptable concentration declared by WHO for drinking water ($< 0.03 \text{ }\mu\text{M}$). These results demonstrated that the polymer could be used to selectively remove Cd^{2+} but not other ions in groundwater, such as Na^+ , K^+ , Mg^{2+} , and Ca^{2+} . Such a high selectivity to toxic Cd^{2+} ions make them distinct from zeolites and ion exchange resins, because the ions possessing similar sizes and charges are equally captured. As presented in Fig. S11, we also detected the decrease in $[\text{Ca}^{2+}]$ in flow-through, suggesting that Ca^{2+} interact with pAA-Cys5. This is reasonable at $[\text{Ca}^{2+}] = 0.5 \text{ mM}$ ($5 \times 10^{-4} \text{ M}$), which is more than one order of magnitude higher than the K_D value of pAA-Cys5 and Ca^{2+} ($\sim 10^{-5} \text{ M}$). As presented in Fig. S12, the interactions of pAA-Cys5 with Na^+ , K^+ , and Mg^{2+} are much weaker. Notably, the interaction of pAA-Cys5 and Ca^{2+} could not be detected by FTIR spectra, which were measured even at higher concentrations ($\sim 10^{-3} \text{ M}$, Figs. 2c and 2d). This finding can be explained by the molar fraction of Cys side chains (5 mol%). Namely, only a small portion of $-\text{COO}^-$ groups contributes to the spectral signals. In fact, the change in the spectral intensity could be detected only in differential spectra even at $[\text{Cd}^{2+}] \sim 10^{-3} \text{ M}$, which is about six magnitudes larger than the K_D value ($\sim 10^{-9} \text{ M}$). Therefore, it seems reasonable that we could not detect the interaction of pAA-Cys5 with Ca^{2+} and other cations spectroscopically.”

In addition, more work is needed to demonstrate a reliable and robust engineering system to evaluate the separating performance of pAA-Cys5 either in the particle or membrane platform. Current results and discussion have not been well-developed because they lack of in-depth analyses and discussion beyond a random engineering demonstration.

We presented “*the separating performance of pAA-Cys5 in the particle platform*” in Fig. 4.

As the reviewer suggested, we performed additional experiments with the unmodified cellulose membrane and silica microparticles without polymers (Fig. 5d, no polymer) and cellulose/chitosan–g–pAA–Cys5 membrane (Fig. 5d, membrane) to examine “*the separating performance of pAA-Cys5 in the membrane platform*”.

We also performed more control experiments, as also suggested by the reviewer 1.

The ITC data of pAA with Ca^{2+} and Cd^{2+} in Supplementary Fig. S3, which clearly indicate that pAA does not interact with Cd^{2+} ions.

For a better clarity, we modified the manuscript as:

“In contrast, the binding affinities of pAA (devoid of cysteine side chains) to Ca^{2+} and Cd^{2+} ions are too low to calculate the K_D values from the ITC data (Fig. S3)”

“In the presence of unmodified cellulose membrane and silica microparticles without polymers, the system could not remove Cd^{2+} , showing a significant leakage from the very first fraction (Fig. 5d).

Even in the absence of functionalized microparticles, the cellulose/chitosan–g–pAA–Cys5 membrane could remove $[\text{Cd}^{2+}]$, but a distinct leaking of Cd^{2+} could be detected already at $V = 60$ mL (open

symbols, Fig. 5d). ... In contrast, in the presence of both...”

Data are shown without error bars, and filled with lots of cartoon demonstrations.

We thank the reviewer for pointing out that the error bars were not presented for some figures. Following the reviewer’s comment, we added the error bars to Fig. 3d, taking the calibration accuracy of Spectroquant® colorimetric assay. The caption to Fig. 3d reads “Error bars indicate the calibration accuracy of Spectroquant® colorimetric assay.”

The data presented in Figure 2 and Figure 3b were already shown with the error bars. We also added the error bars to Figs. 4c and 5d. The errors in X axis originate from the fraction volume, whereas those in Y axis from the calibration accuracy of Spectroquant® colorimetric assay.

While these cartoon demonstrations are very helpful to visualize how the material system look like, there is a need for SEM (and/or TEM) of the pAA-Cys5-drafted silica nanoparticles and membranes used in the demonstration.

The SEM images of silica particles are shown in the new Supplementary Information Fig. S22a. As

lipid membrane cannot survive under the experimental condition for SEM, we present the three-dimensional image of *pAA-Cys5-drafted silica nanoparticles* (precisely, pAA-Cys5-coated silica microparticles) reconstructed from the confocal microscopy images. For a better clarity, scale bars are given in both images. For the confocal imaging, we labeled the polymer chains with Cy3 (Fig. S22b).

This was clarified in the revised manuscript:

“The scanning electron microscopy image of the particle surface and the three-dimensional image of pAA-Cys5-coated microparticles reconstructed from the confocal microscopy images are presented in Fig. S22.”

Also, it is unclear why the membrane-only system shows a cadmium leakage at 60-mL feed water volume - whether this cadmium leakage is due to the membrane integrity or the low loading capacity of pAA-Cys5 on the membrane. If the latter is true, one would be able to address that by increasing the pAA-Cys5 loading on the membrane structure, and systematically study the impacts of pAA-Cys5 loading/density on the Cd ion uptake.

As the reviewer commented, the leakage of Cd^{2+} in the membrane-only system can be explained by the low loading capacity of pAA-Cys5 on the membrane. In our experimental system, we grafted 50 mg chitosan-g-pAA-Cys5 (corresponding to 2.0 μmol of pAA-Cys5) and 63 mg of pAA-Cys5-DOPE (corresponding to 8.9 μmol of pAA-Cys5). This cannot explain the onset of leakage at 60 mL for the membrane-only system and 300 mL for the membrane-particle systems. Thus, it is plausible that the leakage is caused by a too high flow rate through the membrane-only systems, 5 mL min⁻¹ (300 mL h⁻¹), which is about 250 times larger than that of particle/molecular sieve system.

To verify this hypothesis, we performed the experiments by immobilizing lower amounts of the chitosan-g-pAA-Cys5 grafted to the cellulose membranes (Fig. S29). The nonlinear dependence of the onset volume of the leakage on the amount of grafted chitosan-g-pAA-Cys5 suggests that the membrane cannot follow the high flow rate.

In the revised manuscript, we explained this point:

“The loading of different amounts of chitosan-g-pAA-Cys5 exhibited a nonlinear dependence of the onset volume of the leakage on the amount of grafted chitosan-g-pAA-Cys5 (Fig. S29). This suggests that the combination of the cellulose/chitosan-g-pAA-Cys5 membrane and the pAA-Cys5-coated particles is necessary, because the membrane alone is not sufficient to capture Cd^{2+} under the practical flow rate used for the water treatment (5 mL min^{-1}).”

While the “membrane + particle” system does show a better performance, we do not understand the main factor that govern such an observation beyond the collective effects of both pAA-Cys5 particles and pAA-Cys5 membranes. Essentially, additional characterization and analyses are essential to elucidate the main factors governing such a performance.

The data presented in the new Fig. 5d and those presented in our previous answer indicate that the membrane system are able to catch Cd^{2+} to some extent, but the membrane system alone is not able to capture Cd^{2+} under a high flow rate (5 mL min^{-1}).

If we operate our system under the condition used for other fixed-bed column systems, the combination of the cellulose/chitosan-g-pAA-Cys5 membrane and the particle system is essential.

Note that the particle-only system using the molecular sieve used for the column system (pore diameter: $0.1 \mu\text{m}$) instead of the membrane system was not examined, because a 45 mm-large molecular sieve (corresponding to the inner diameter of the integrative system) was not practically available.

We believe our response to the previous comment provides a clear answer to this point.

“The loading of different amounts of chitosan-g-pAA-Cys5 exhibited a nonlinear dependence of the onset volume of the leakage on the amount of grafted chitosan-g-pAA-Cys5 (Fig. S29). This suggests that the combination of the cellulose/chitosan-g-pAA-Cys5 membrane and the pAA-Cys5-coated particles is necessary, because the membrane alone is not sufficient to capture Cd^{2+} under the practical flow rate used for the water treatment (5 mL min^{-1}).”

Minor points:

- The term “hyperconfinement” appears to be misused in this work, where basically, the particles are packed in a small volume, and thus polymer chains can be densely packed. Yes, by drafting the copolymers onto silica nanoparticles, the authors can increase the loading (or packing density) of the material per a unit volume, however, stating that the polymers are hyperconfined without a proof of evidence appears to be somewhat overreaching.

We disagree, because the increasing the *packing density material per a unit volume cannot be achieved unless the polymers are grafted the surface of silica microparticles*. We believe to call it as “hyper-confinement” is not *overreaching*, because the particle-membrane system showed a better performance.

-While it is understandable that the regeneration of the material system and release of Cd ion can be challenging and might be within a new scope of study, it is not convincible to state that “the heavy metal and silica particles can be recovered by burning organic materials”. Given the several steps involved in the synthetic process to achieve such a copolymeric system, which may involve several purification steps, burning the organic materials to recover the traces of heavy metal and silica particles might not be helpful in our intensive efforts to minimalize global carbon footprint.

We thank the reviewer for raising this key point. In fact, a similar comment came from the reviewer 1, too (comment 22).

Our statement is based on the differential melting/boiling points of organic compounds, metals and silica. The melting point and the boiling point of silica are 1710 °C (1984 K) and 2230 °C (2504 K), respectively. The corresponding values for Cd are 321 °C (595 K) and 765 °C (1039 K), respectively. As organic materials are supposed to burned out at 600 °C (*RSC Adv.* **5**, 42572-42579 (2015)), we assume that the heating of the systems to 600 °C results in solid silica and melted, liquidous Cd. To avoid the confusion, we added explanatory text and clarify that this is one possibility expected from the materials’ properties.

We have confirmed that the polymers can be regenerated by eluting EDTA solution, and the data are presented in Fig. S31. Nevertheless, it should be noted that this regeneration procedure enables to reuse polymers, but Cd²⁺ ions are just transferred to another aqueous solution (flow through).

The advantages and disadvantages of these two recovery strategies, “burning of organic materials” vs. “the recovery of polymers with EDTA”, are discussed in the revised manuscript more explicitly.

“The regeneration and recovery of the materials can potentially be achieved by the following two strategies. In the first strategy, the polymers can be regenerated by cancelling the chelate complex by adding a competitor, such as EDTA. As presented in Fig. S31, the continuous treatment with the buffer containing 10 mM EDTA for 200 min resulted in the recovery of function by 83%. Although this suggests the recyclability of the materials, this treatment is to simply concentrate Cd^{2+} ions from one aqueous phase (polluted water model) to another (EDTA buffer). In the second strategy, the heavy metal and silica particles can be recovered by taking the difference in temperature conditions for decomposition into account. For example, the organic compounds can be removed and the metal (Cd) can be recovered in a fluid phase at 600 °C, which is well below the melting temperature of silica (1710 °C). Although the potential recovery of heavy metal can add a unique advantage to our materials over zeolites and other porous inorganic materials, this approach might not be helpful for the minimization of global carbon footprints.”

- The flow rate of 0.005 liter/min demonstrated in the flow-through membrane system is impractically low.

Previously, we fabricated a flow-through biochemical microreactor by packaging sarcoplasmic reticulum membrane deposited on silica microparticle with a diameter of 10 μm in a chromatography column with an inner diameter of 10 mm [Tutus, et al., *Adv. Funct. Mater.* **22**, 4873–4878 (2012)]. For the first prototype (Figure 4), we selected the same particle size and the column size, which resulted in the flow rate of 1.5 mL/h, which was about one half of the previous study.

In general, the technical limit of the flow rate is defined by the back pressure P , which follows:

$$P \propto \frac{L \times F}{D^2 \times ID^2}.$$

L is the length of the path (packed beads), F the flux, D the particle diameter and ID is the inner diameter

of the column. The use of larger particles decreases the pressure if L is kept constant, e.g. the use of 20 μm -large particles reduces the P by a factor of 4. However, in order to keep the same surface area, one needs 2 times larger bed volume and hence the length L .

Therefore, we concluded that the prototype was useful to demonstrate the proof of principle but not suited for achieving a realistic flow rate comparable to a fixed-bed column system [*ChemBioEng Rev.* **5**, 173-179 (2018)].

Using the membrane-particle system, we successfully increased the flow rate:

“Notably, the new system could be operated at a flow rate of 5 mL min⁻¹, **which is 250 times higher than the flow rate used for the previously described prototype** (Fig. 4).”
as described in the original manuscript.

In the revised manuscript, we added more precise explanation:

“... in practice. This extremely low flow rate is caused by a high back pressure P , which scales with:

$$P \propto \frac{L \times F}{D^2 \times ID^2}$$

L is the length of the path (packed beads), F the flux, D the particle diameter and ID is the inner diameter of the column. The use of larger particles decreases the pressure if L is kept constant, e.g. the use of 20 μm -large particles reduces the P by a factor of 4. However, in order to keep the same surface area, one needs 2 times larger bed volume and hence the length L . Therefore, we concluded that the prototype was useful to demonstrate the proof of principle but not suited for achieving a realistic flow rate comparable to a fixed-bed column system.⁴⁶”

Reviewer #3 (Remarks to the Author):

The manuscript presents a biomimetic approach to address the global demand for clean water, specifically focusing on the design and application of phytochelatin-inspired copolymers for efficient removal of hazardous heavy metal ions from contaminated waters. The synthesized copolymers, precisely decorated with carboxylate and thiolate moieties, designated as pAA–Cys5, demonstrate exceptional Cd²⁺ ion-capturing capacity in presence of Ca²⁺. By employing detailed characterization techniques and comparing the activity with various synthesized variants, the manuscript elucidates the molecular-level mechanisms underlying polymer-metal ion complex formation. The application of these bio-inspired copolymers in flow-through systems has been demonstrated as highly effective, selectively removing Cd²⁺ ions from heavy metal ion contaminated water and achieving levels suitable for meeting drinking water standards. The study not only provides valuable insights into the development of water treatment materials but also showcases the potential of biomimetic strategies for environmental remediation.

Technical questions and additional discussion required:

1. Figure 5: It would be valuable for readers if the authors could consider including permeability values for water in this study. This addition would enhance the comparison of the material's performance with other membranes, providing a more comprehensive understanding of its characteristics.

We thank the reviewer for raising this important point.

In flow-through systems, the water permeability is generally given by linear velocity, which is flow rate divided by cross-sectional area. The linear velocity of the particle/molecular sieve system (Fig. 4) was 0.02 m h⁻¹, whereas that of the membrane-particle system (Fig. 5) was ten times larger, 0.2 m h⁻¹. The linear velocity of the membrane-particle system is comparable to those of other fixed-bed column systems, ~ 0.1–1 m h⁻¹ [*ChemBioEng Rev.* **5**, 173-179 (2018)].

Following the reviewer's suggestion, we modified the manuscript as:

“The flow rate of our system (300 mL h⁻¹) is comparable to those of fixed-bed column systems, ~ 10 – 1000 mL h⁻¹.⁴⁶ The linear velocity, a system size-independent index of water permeability, of our system was 0.2 m h⁻¹, which is also comparable to those of conventional systems, ~ 0.1 – 1 m h⁻¹.⁴⁶”

2. It would be beneficial if the authors could include information about the regeneration or cleaning process of the materials, along with insights into the potential loss of adsorption quality over multiple cycles in the flow-through reactor. This addition would contribute to a more comprehensive assessment of the material's cost-effective performance and suitability for potential commercial applications.

We thank the reviewer for raising this important point. In fact, a similar comment was also raised by the reviewers 1 and 2.

We have confirmed that the polymers can be regenerated by eluting EDTA solution, and the data are presented in Fig. S31. Nevertheless, it should be noted that this regeneration procedure enables to reuse polymers, but Cd^{2+} ions are just transferred to another aqueous solution (flow through). In the revised manuscript, we explained these additional data and discussed about the advantage and disadvantage of this approach.

“The regeneration and recovery of the materials can potentially be achieved by the following two strategies. In the first strategy, the polymers can be regenerated by cancelling the chelate complex by adding a competitor, such as EDTA. As presented in Fig. S31, the continuous treatment with the buffer containing 10 mM EDTA for 200 min resulted in the recovery of function by 83%. Although this suggests the recyclability of the materials, this treatment is to simply concentrate Cd^{2+} ions from one aqueous phase (polluted water model) to another (EDTA buffer).”

3. While the selectivity of pAA–Cys5 for Cd^{2+} ions in the presence of Ca^{2+} ions is highlighted, it would significantly enhance the manuscript's quality if the authors could also demonstrate the selectivity of these materials for other toxic heavy metal ions commonly found in heavy metal ions contaminated environmental water or industrial wastewater. This broader assessment would provide a more comprehensive understanding of the material's applicability across various water treatment conditions.

We thank the reviewer for raising this point. As suggested by the reviewer, we performed additional ITC experiments to *demonstrate the selectivity of these materials for other toxic heavy metal ions commonly found in heavy metal ions contaminated environmental water or industrial wastewater.* We consulted a recent account in the field of water science [*npj Clean Water* 4, 36 (2021)] and selected

Hg²⁺ as a representative heavy metal ion interacting with phytochelatin [*Plant Cell Environ.* **34**, 778-791 (2011)]. The data shown in Fig. S32 indicate that the dissociation constant of pAA–Cys5 to Hg²⁺ ($K_D = 7.1 \times 10^{-9}$ M per molecule) is in the same order of magnitude as that to Cd²⁺.

For the clarity, we added Supplementary Fig. S32 and explanatory text block as follows.

“Intriguingly, the dissociation constant of pAA–Cys5 to Hg²⁺ ($K_D = 7.1 \times 10^{-9}$ M per molecule) is in the same order of magnitude as that to Cd²⁺ (Fig. S32). As Hg²⁺ was reported to be captured by plant phytochelatin,⁴⁸ this result suggests that pAA–Cys5 is able to capture Hg²⁺, as plant phytochelatin does.”

4. Additionally, in connection with the aforementioned point, incorporating selectivity data for heavy metal ions in the presence of mixed ions, including a mixture of more abundant mono and divalent metal ions along with heavy metal ions to simulate real-world conditions, would not only demonstrate the efficiency of this material but also significantly broaden its potential scope in practical applications.

We thank the reviewer for raising this important point. In fact, a similar comment was also raised by the reviewers 1 and 2.

To directly answer the comment “*selectivity data for heavy metal ions in the presence of mixed ions, including a mixture of more abundant mono and divalent metal ions along with heavy metal ions to simulate real-world conditions*”, we performed additional experiments in the coexistence of [K] = 10 µM, [Na] = 1000 µM, [K] = 200 µM, [Mg] = 500 µM, and [Ca] = 500 µM, which is a realistic model of the ions in the ground water. The obtained data are presented in Supplementary Fig. S30.

In the revised manuscript, we clarified this point:

“...within 1 h. The Cd²⁺ removal capacity of our system was further examined by treating the simulated wastewater containing [Cd²⁺] = 0.01 mM (10 μM), [Na⁺] = 1 mM, [K⁺] = 0.2 mM, [Mg²⁺] = 0.5 mM, and [Ca²⁺] = 0.5 mM. The [Cd²⁺] level remained below the detection limit of the colorimetric assay up to 135 mL at a flow rate of 5 mL min⁻¹ (Fig. S30).”

We also added the following text and Fig. S12 in the revised manuscript:

“To test if pAA–Cys5 can selectively capture Cd²⁺, we incubated pAA–Cys5 solutions (10⁻⁴, 10⁻³, 10⁻², and 10⁻¹ mg mL⁻¹) with a buffer containing [Cd²⁺] = 0.01 mM (10 μM), [Na⁺] = 1 mM, [K⁺] = 0.2 mM, [Mg²⁺] = 0.5 mM, and [Ca²⁺] = 0.5 mM, which mimic the concentration levels in wastewater, respectively³⁴ Fig. 3d shows [Cd²⁺] in flow-through, indicating that pAA–Cys5 can capture Cd²⁺ even in the presence of a large excess of abundant mono- and divalent metal ions in ground water. When using 0.1 mg mL⁻¹ of pAA–Cys5, [Cd²⁺] in flow-through was lower than the acceptable concentration declared by WHO for drinking water (< 0.03 μM). These results demonstrated that the polymer could be used to selectively remove Cd²⁺ but not other ions in groundwater, such as Na⁺, K⁺, Mg²⁺, and Ca²⁺. Such a high selectivity to toxic Cd²⁺ ions make them distinct from zeolites and ion exchange resins, because the ions possessing similar sizes and charges are equally captured. As presented in Fig. S11, we also detected the decrease in [Ca²⁺] in flow-through, suggesting that Ca²⁺ interact with pAA–Cys5. This is reasonable at [Ca²⁺] = 0.5 mM (5 × 10⁻⁴ M), which is more than one order of magnitude higher than the K_D value of pAA–Cys5 and Ca²⁺ (~ 10⁻⁵ M). As presented in Fig. S12, the interactions of pAA–Cys5 with Na⁺, K⁺, and Mg²⁺ are much weaker. Notably, the interaction of pAA–Cys5 and Ca²⁺ could not be detected by FTIR spectra, which were measured even at higher concentrations (~ 10⁻³ M, Figs. 2c and 2d). This finding can be explained by the molar fraction of Cys side chains (5 mol%). Namely, only a small portion of –COO⁻ groups contribute to the spectral signals. In fact, the change in the spectral intensity could be detected only in differential spectra even at [Cd²⁺] ~ 10⁻³ M, which is about six magnitudes larger than the K_D value (~ 10⁻⁹ M). Therefore, it seems reasonable that we could not detect the interaction of pAA–Cys5 with Ca²⁺ and other cations spectroscopically.”

REVIEWERS' COMMENTS

Reviewer #1 (Remarks to the Author):

Thanks to the authors for taking the time to make modifications. For me, the work is now acceptable.

Reviewer #3 (Remarks to the Author):

In the revised manuscript, the authors have meticulously conducted numerous experiments and thoroughly addressed the concerns raised by this reviewer. They have provided detailed responses, demonstrating a comprehensive understanding of the issues with well-supported experiments. The additional data and analyses significantly strengthen the overall findings, ensuring that the manuscript meets the high standards expected for publication.

[Note from the Editor: Reviewer #3 was asked to look also over the response given to Reviewer #2 and thinks that the authors have addressed all the concerns, either through detailed explanations or by providing additional supporting experiments.]